# From Sparse to Dense: Spatio-Temporal Fusion for Multi-View 3D Human Pose Estimation with DenseWarper

**Ling Li[1,†], Changjie Chen[2,†], Yuyan Wang[5], Jiaqing Lyu[1], Kenglun Chang[3], Yiyun Chen[4], Zhidong Deng [1,\*]**

[1]Department of Computer Science, THUAI, BNRist, Tsinghua University, Beijing, China
[2]Dalian University of Technology, Dalian, China
[3]Apple, Beijing, China
[4]Hong Kong University of Science and Technology (Guang Zhou), Guang Zhou, China
[5]University of Manchester, Manchester, UK
[†]Equal contribution
[\*]Corresponding author: michael@tsinghua.edu.cn

## Abstract

In multi-view 3D human pose estimation, models typically rely on images captured simultaneously from different camera views to predict a pose at a specific moment. While providing accurate spatial information, this traditional approach often overlooks the rich temporal dependencies between adjacent frames. We propose a novel 3D human pose estimation input method: the sparse interleaved input to address this. This method leverages images captured from different camera views at various time points (e.g., View 1 at time $t$ and View 2 at time $t + \delta$), allowing our model to capture rich spatio-temporal information and effectively boost performance. More importantly, this approach offers two key advantages: First, it can theoretically increase the output pose frame rate by N times with N cameras, thereby breaking through single-view frame rate limitations and enhancing the temporal resolution of the production. Second, using a sparse subset of available frames, our method can reduce data redundancy and simultaneously achieve better performance. We introduce the DenseWarper model, which leverages epipolar geometry for efficient spatio-temporal heatmap exchange. We conducted extensive experiments on the Human3.6M and MPI-INF-3DHP datasets. Results demonstrate that our method, utilizing only sparse interleaved images as input, outperforms traditional dense multi-view input approaches and achieves state-of-the-art performance. The source code for this work is available at https://github.com/lingli1724/DenseWarper-ICLR2026.

## 1 Introduction

3D Human Pose Estimation (3DHPE) (Li et al., 2023; Baumgartner & Klatt, 2023; Bridgeman et al., 2019; Qiu et al., 2019b; Zheng et al., 2020; Li et al., 2025; 2024a) has broad applications in fields such as dance synthesis (Wang et al., 2024a; Yin et al., 2024; Wang et al., 2024b), action recognition (Karim et al., 2024; Manakitsa et al., 2024; Li et al., 2024b), and virtual reality (Lampropoulos & Kinshuk, 2024; de Lurdes Calisto & Sarkar, 2024). Especially in multi-view settings, utilizing images captured by multiple synchronized cameras can provide more accurate and stable pose estimation results than single-view methods (Hyla, 2016; Bridgeman et al., 2019; Zheng et al., 2021a; Pavllo et al., 2019b). However, existing methods commonly adopt a single-moment dense input paradigm, where the model must receive synchronized images from all views at each time step to predict the pose. Although this approach provides sufficient spatial information, its single-moment input structure presents three key bottlenecks: redundant computational overhead, insufficient utilization of temporal information (Xu et al., 2024b; Xie et al., 2024), and the inability to break through camera frame rate limitations. This inefficient input mode leads to unnecessary computational waste and temporal information discontinuity (Han et al., 2024; Liu et al., 2020a).

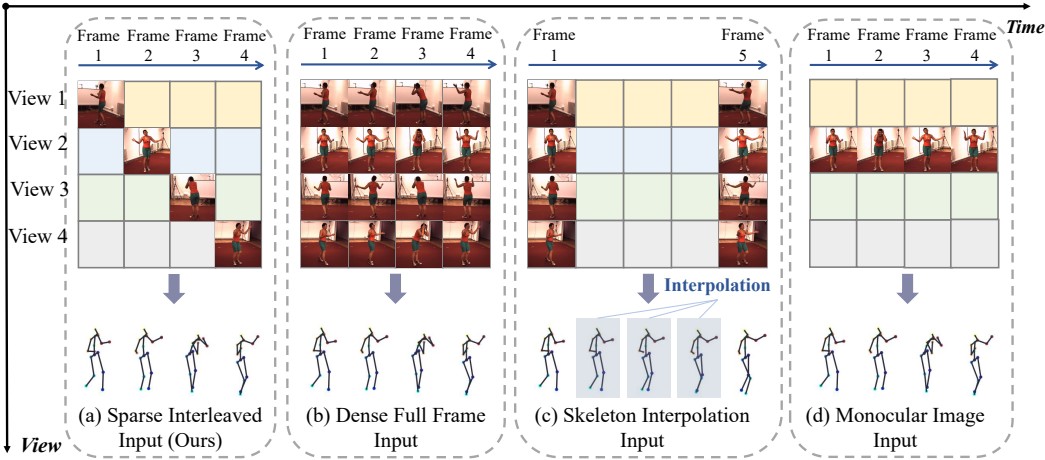

**Figure 1:** Common approaches for 3D multi-view pose estimation. (a) Our proposed sparse interleaved input, where each view selects a single temporally interleaved image as input to leverage spatio-temporal information across views fully; (b) illustration of dense, full-frame multi-view input; (c) keypoint interpolation input, which enhances the output frame rate; and (d) illustration of single-view image input.

We propose a novel 3D human pose estimation input paradigm to solve this challenge: sparse interleaved input. This method breaks the limitations of traditional synchronous input by cleverly using images captured from different camera views at various time points as input. More specifically, Camera 1 captures an image at time $t$, followed by Camera 2 at $t + \delta$, and so forth, with the sequence culminating in Camera $N$ at $t + (N - 1) \times \delta$. This innovative method brings two core advantages. First, it efficiently utilizes spatio-temporal information and can be mathematically viewed as a joint spatio-temporal sampling. The method cleverly leverages the temporal phase differences of multi-view inputs to reconstruct a higher-frequency pose output signal in the temporal dimension. Second, it fundamentally breaks through the single-camera frame rate limitation. Specifically, for $N$ fixed-frame-rate cameras with a fixed sampling rate of $F$, the sampling interval is $N \times \delta$. Due to the interleaved sampling between views, the pose output interval is $\delta$, which raises the camera's effective sampling rate to $N \times F$, opening up a new path for 3D tasks requiring high temporal resolution. To solve the computational latency problem in practical applications, we adopt a sliding window optimization strategy, allowing the model to sample and process data instantly without waiting for all cameras to complete their interleaved sampling, achieving efficient and real-time processing.

To achieve this goal, we designed an end-to-end framework named DenseWarper. This framework can efficiently convert sparse interleaved inputs into dense outputs with high spatio-temporal consistency. DenseWarper includes two core modules: spatial rectification and temporal fusion modules. First, we use a spatial transformation module based on epipolar geometry to spatially rectify and fuse 2D heatmaps, generating a preliminary dense heatmap. Next, we use a temporal fusion module based on deformable convolution to learn and complete the temporal information in the heatmaps implicitly. Finally, a precise 3D pose can be obtained through triangulation.

Our main contributions are as follows:

Pioneering Task Paradigm: We are the first to propose and define the 3D pose estimation task based on sparse interleaved multi-view input. We provide a novel paradigm for efficiently utilizing spatio-temporal information and potentially inspiring research in other multi-view 3D perception tasks.

DenseWarper Framework: We designed DenseWarper, which uses an innovative spatio-temporal fusion mechanism to convert sparse interleaved inputs into dense pose outputs with high spatio-temporal consistency. It breaks the frame rate limitations of traditional 3D tasks and, combined with the sliding window strategy, achieves low-latency processing.

Rigorous Experimental Validation and Benchmark Establishment: We conducted comprehensive experiments on the Human3.6M (Ionescu et al., 2013) and MPI-INF-3DHP (Mehta et al., 2017) datasets. We reproduced numerous state-of-the-art 3D pose estimation algorithms and tested their performance on the MPI-INF-3DHP dataset. Using a unified benchmark for the 2D part, we fill a gap in testing benchmarks on MPI-INF-3DHP, providing a rigorous and reliable reference for algorithms in this field.

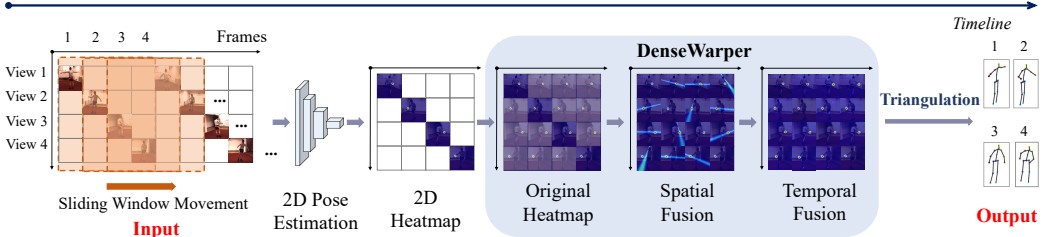

**Figure 2:** Overview of the DenseWarper architecture. A sliding window is used to sample sparse interleaved images, with a 2D pose estimation model generating initial heatmaps for each view. Missing information is filled to create uncorrected heatmaps. These are then spatially fused and corrected using an epipolar geometry-based method, yielding a spatially fused heatmap. Deformable convolutions are then applied for temporal fusion. Finally, the resulting spatiotemporally enriched heatmap is processed via triangulation to obtain accurate 3D keypoints.

## 2 PROBLEM FORMULATION

**Sparse Interleaved Multi-view Images.** In this study, we introduce the concept of **"Sparse Interleaved Multi-View Images"** to describe a sequence of sparsely sampled images from multiple viewpoints in a multi-view setting. Specifically, we define them as a collection of sparsely sampled images from multiple viewpoints, represented as:

$$\mathbf{D} = \{\mathbf{I}_i\}_{i=1}^{\lfloor \frac{N}{M} \rfloor}, \tag{1}$$

where $M$ is the number of viewpoints $V = \{V_1, V_2, \ldots, V_M\}$, $N$ is the total number of frames, and $i$ is the index of the input group. Each group $\mathbf{I}_i$ contains one image from each viewpoint, ranging from the $\{M \cdot (i-1) + 1\}$-th frame of viewpoint $V_1$ to the $\{M \cdot i\}$-th frame of viewpoint $V_M$. Here, the $i$-th group is denoted as:

$$\mathbf{I}_i = \left\{ I_{V_1}^{M \cdot (i-1)+1}, I_{V_2}^{M \cdot (i-1)+2}, \ldots, I_{V_M}^{M \cdot i} \right\}, \tag{2}$$

where $I_{V_j}^{M \cdot (i-1)+j} \in \mathbb{R}^{H \times W \times 3}$ represents the image from viewpoint $V_j$ with frame number $M \cdot (i-1) + j$. This structure ensures each time frame contains only one image from a specific view, creating an interleaved sparse input.

Taking $M = 4$ as an example, the sparse interleaved inputs for the first two frame groups are $\mathbf{I_1} = \left\{ I_{V_1}^1, I_{V_2}^2, I_{V_3}^3, I_{V_4}^4 \right\}$ and $\mathbf{I_2} = \left\{ I_{V_1}^5, I_{V_2}^6, I_{V_3}^7, I_{V_4}^8 \right\}$.

**Target for Modeling.** Traditional multi-view 3D human pose estimation tasks rely on densely synchronized multi-view images as input. In contrast, our approach leverages sparse interleaved multi-view image sequences $\mathbf{D}$.

The objective of our model is to predict a set of 3D skeletons $S = \{S_1, S_2, \ldots, S_N\} \in \mathbb{R}^{N \times J \times 3}$, where each $S_i \in \mathbb{R}^{J \times 3}$ represents the 3D pose coordinates at the $i$-th frame for $J$ keypoints. We aim to achieve pose estimation performance comparable to dense full-frame input. Formally, we define the mapping as follows:

$$f : \mathcal{T}(\mathcal{D}, \Phi, \phi) = S, \tag{3}$$

where $\Phi$ denotes the 3D pose estimation model designed for sparse interleaved inputs, $\phi$ represents the camera parameters, $S \in \mathbb{R}^{N \times J \times 3}$ is the resulting set of 3D skeletons, and $J$ represents the number of keypoints.

For each specific interleaved input group $\mathbf{I}_i$, such as $\mathbf{I}_1 = \{ I_{V_1}^1, I_{V_2}^2, I_{V_3}^3, I_{V_4}^4 \}$, the model produces a corresponding set of 3D skeletons $\{S_1, S_2, S_3, S_4\}$, where each skeleton $S_n$ corresponds to the pose at the $n$-th frame. This can be expressed mathematically as:

$$f : \mathcal{T}(\{\mathbf{I}_{V_i}^{M \cdot (i-1)+j}\}_{j=1}^M, \Phi, \phi) = \{S_{M \cdot (i-1)+1}, \ldots, S_{M \cdot i}\}, \tag{4}$$

where $I_{V_i}^{M \cdot (i-1)+j} \in \mathbb{R}^{H \times W \times 3}$ is the image from view $V_j$ at the frame number $M \cdot (i-1) + j$, and $S_{M \cdot (i-1)+j}$ represents the predicted 3D pose coordinates for that time frame.

**Sliding Window Mechanism.** In conventional methods, a new input group $\mathbf{I}_i$ must wait for all $M$ views to complete sampling. For instance, with $M = 4$, the second group $\mathbf{I}_2 = (I_{V_1}^5, I_{V_2}^6, I_{V_3}^7, I_{V_4}^8)$ cannot be processed until $I_{V_4}^8$ is available.

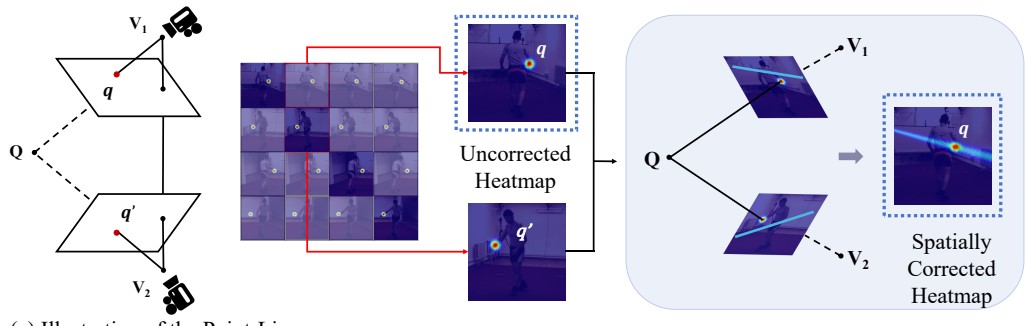

(a) Illustration of the Point-Line
Correspondence in Two Views

(b) Epipolar Geometry-based Spatial Heatmap Fusion

**Figure 3:** Epipolar geometry-based spatial heatmap fusion architecture. (a) Geometric interpretation of the point-line relationship for keypoints across different views; (b) the pipeline for spatial heatmap fusion based on epipolar geometry. For an inaccurate heatmap point $q$, we use accurate points $q'$ from other views to correct it. First, we compute the corresponding epipolar lines in the other two heatmaps. Then, we identify the maximum response along the line associated with $q$ and add these values to the original response at $q$ in its heatmap. This process yields a spatially corrected heatmap. In the figure, non-diagonal heatmaps with masking represent the target heatmaps for correction, all processed according to this method.

The sliding window method allows immediate processing once any view finishes sampling by reusing previously computed heatmaps. For example, after $V_1$ samples $I_{V_1}^5$, a new input can be formed as:

$$\mathbf{I}_2' = \{I_{V_2}^2, I_{V_3}^3, I_{V_4}^4, I_{V_1}^5\},$$

where $I_{V_2}^2$, $I_{V_3}^3$, $I_{V_4}^4$ are already processed and cached.

This mechanism enables real-time incremental processing of sparse interleaved inputs without waiting for all views. A caching mechanism simultaneously allows for the reuse of computed heatmaps, thereby reducing latency and computational cost while maintaining estimation accuracy.

## 3 DENSEWARPER

Figure 2 illustrates our overall network structure, where our core module, DenseWarper, consists of two main components. First, the sparse interleaved heatmaps are input into a spatial fusion module based on epipolar geometry, producing an initially rectified dense 2D heatmap. Next, these initially rectified heatmaps are processed through a temporal correction module, the warping module, which integrates pose information from time frames to generate a dense heatmap with enhanced spatio-temporal consistency. Finally, we utilize the Triangulation method (Iskakov et al., 2019b; Remelli et al., 2020b) to spatially reconstruct all rectified heatmaps, thereby obtaining the 3D human pose. Section 3.1 introduces the concepts related to epipolar geometry and multi-view spatial heatmap fusion; Section 3.2 discusses the temporal fusion process using Warper.

### 3.1 HEATMAP FUSION WITH EPIPOLAR GEOMETRY

#### 3.1.1 BASIC CONCEPT OF EPIPOLAR GEOMETRY

**Epipolar Geometry.** It describes the geometric relationship between the corresponding points that two cameras observe in a 3D scene. Readers can refer to (Hartley & Zisserman, 2003). This relationship is established by the epipolar constraint and the fundamental matrix (He et al., 2020; Xu & Zhang, 2013; Zhang, 1998), which are widely used in tasks such as multi-view 3D reconstruction.

Let a 3D point $\mathbf{Q} \in \mathbb{R}^4$, with its fourth dimension set to 1, be projected onto the image planes of two cameras $V_1$ and $V_2$, with 2D projections $\mathbf{q} \in \mathbb{R}^3$ and $\mathbf{q}' \in \mathbb{R}^3$ respectively, where the third dimension of them is 1. The projection matrices $\mathbf{P}$ and $\mathbf{P}'$ map $\mathbf{Q}$ onto each image plane can be denoted as:

$$\mathbf{q} = \mathbf{P}\mathbf{Q}, \quad \mathbf{q}' = \mathbf{P}'\mathbf{Q}. \tag{5}$$

According to epipolar geometry, the corresponding points $\mathbf{q}$ and $\mathbf{q}'$ satisfy the epipolar constraint, defined by a fundamental matrix $\mathbf{F} \in \mathbb{R}^{3\times3}$ as follows:

$$\mathbf{q}'^\top \mathbf{F}\mathbf{q} = 0. \tag{6}$$

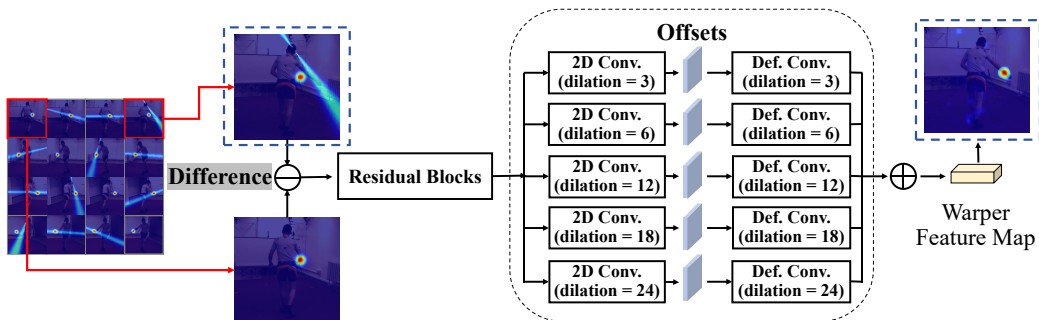

**Figure 4:** The structure of the temporal fusion module (Warper). We perform temporal correction based on the initial corrected heatmaps obtained from multi-view spatial fusion. For each heatmap in a target time frame (i.e., non-diagonal heatmaps in the figure), we compute its difference with the corresponding accurate heatmap in the same view (the diagonal heatmap) and apply a temporal pose feature learning module to correct the heatmaps along the temporal dimension further. We feed the computed differences into a stack of $3 \times 3$ residual blocks, followed by five $3 \times 3$ convolutional layers with dilation rates $d \in \{3, 6, 12, 18, 24\}$. Each convolutional layer predicts a set of five offsets $o^{(d)}(p_n)$ for each pixel location $p_n$, which are used to rewarp pose heatmap $B$. The five rewarped heatmaps are then summed, and the resulting tensor is used to predict the target heatmap.

This constraint indicates that, given a point $\mathbf{q}$ in the first view, its corresponding point $\mathbf{q}'$ must lie on the epipolar line $\mathbf{l}' = \mathbf{Fq}$ in the second view, thus reducing the correspondence search from a 2D region to a 1D line.

**Application in Multi-View Fusion.** In the DenseWarper model, we aim to use the Sampson distance to correct the heatmaps of missing time frames within each view, thereby fully leveraging the information across different views. The appendix explains our framework's epipolar geometry principles and Sampson distance metric.

### 3.1.2 HEATMAP FUSION

As shown in Figure 2, we can obtain the corresponding heatmaps based on the RGB image input (Gu, 2022; Zhang et al., 2021; Li et al., 2020). We define the first set of sparse interleaved heatmaps as $\{\mathbf{H}_{V_1}^1(x), \mathbf{H}_{V_2}^2(x), \dots, \mathbf{H}_{V_M}^M(x)\}$. For ease of discussion, all subsequent formulations will be based on the first heatmap set, where $\mathbf{H}_{V_j}^n(x)$ represents the heatmap for view $V_j$ in the $n$-th frame at location $x$, where $j \in [1, M]$. To supplement the missing information, we first replicate each heatmap $\mathbf{H}_{V_n}^n(x)$ across the disappeared frames for each view, resulting in an expanded set $\mathbf{H}$ of replicated heatmaps.

Thus, the expanded set $\mathbf{H}$ of heatmaps is formulated as:

$$\mathbf{H} = \{\{\mathbf{H}_{V_1}^1(x), \mathbf{H}_{V_1}^1(x), \dots, \mathbf{H}_{V_1}^1(x)\}, \{\mathbf{H}_{V_2}^2(x), \mathbf{H}_{V_2}^2(x),$$
$$\dots, \mathbf{H}_{V_2}^2(x)\}, \dots, \{\mathbf{H}_{V_M}^M(x), \mathbf{H}_{V_M}^M(x), \dots, \mathbf{H}_{V_M}^M(x)\}\} \tag{7}$$

where each original heatmap $\mathbf{H}_{V_n}^n(x)$ is replicated $M - 1$ times to fill the blank in view $V_n$.

Due to incomplete alignment, the replicated heatmaps introduce spatial-temporal offsets, which can be corrected using epipolar geometry. This approach aligns each replicated heatmap with precise spatial information from other views. Since features at low-probability locations along the epipolar line contribute minimally to cross-view fusion, exact correspondences between views are unnecessary. Instead, we select the maximum probability point along the epipolar line as the matching point, which is a reasonable simplification, as accurate correspondences typically yield the highest response. For instance, as shown in Figure 3, for each position $x$, we compute the epipolar lines in the other views and fuse the maximum responses along these lines with the response at $x$.

Specifically, for each replicated heatmap $\mathbf{H}_v^n(x)$ in view $v$, we use the corresponding heatmap from another view $u$ ($u \neq v$) to provide the correct spatial information. The correction process for each heatmap $\mathbf{H}_v^n(x)$ in view $v$ at the $n$-th frame is defined by:

$$\hat{\mathbf{H}}_v^n(x) = \lambda \mathbf{H}_v^n(x) + \frac{(1 - \lambda)}{M} \sum_{u=1}^{M} \max_{x' \in \mathbf{p}^u(x)} \mathbf{H}_u^n(x'), \tag{8}$$

**Table 1:** MPJPE Comparison with state-of-art pose estimation methods on Human3.6M (mm) using ground-truth and detected 2Dposes. Input types: Single-view (Single), Multi-view full-frame (Full), Multi-view interpolated (Interp). Best in bold.

| METHOD | INPUT | ACTIONS | | | | | | | | | | | | | | | AVG |
|---|---|---|---|---|---|---|---|---|---|---|---|---|---|---|---|---|---|
| | | Dir. | Disc. | Eat. | Greet. | Phone. | Photo. | Pose. | Pur. | Sit. | SitD. | Smoke. | Wait. | Walk. | WalkD. | WalkT. | |
| *2D—Ground Truth (GT)* | | | | | | | | | | | | | | | | | |
| GLA-GCN (T=243) (Yu et al., 2023b) | Single | 26.6 | 27.2 | 29.2 | 25.4 | 28.2 | 31.7 | 29.5 | 27.0 | 37.8 | 40.0 | 29.9 | 27.0 | 20.5 | 27.3 | 20.8 | 28.5 |
| KTP-Former (T=243) (Peng et al., 2024) | Single | **22.7** | 23.4 | 21.8 | 22.5 | 24.2 | 29.9 | 25.7 | 22.9 | 30.3 | 36.9 | 24.4 | 23.3 | **17.3** | 24.3 | **18.2** | 24.5 |
| Adafuse (Zhang et al., 2021) | Full | 26.3 | 25.4 | 22.4 | 23.9 | 22.9 | 22.6 | 24.1 | 24.4 | 23.7 | 21.6 | 24.0 | 23.9 | 23.1 | 23.9 | 22.8 | 23.7 |
| Adafuse + MCC (Su et al., 2021) | Interp | 26.0 | 25.6 | 22.2 | 23.6 | 22.3 | 23.9 | 23.8 | 24.3 | 24.4 | 23.3 | 24.4 | 23.9 | 22.3 | 24.9 | 22.4 | 23.8 |
| Adafuse + SLERP (Chen et al., 2022) | Interp | 26.1 | 25.2 | 22.3 | 23.6 | 22.7 | 22.4 | 23.8 | 24.3 | 23.6 | 21.4 | 23.8 | 23.8 | 22.9 | 23.7 | 22.5 | 23.5 |
| Adafuse | Sparse | 27.3 | 27.4 | 23.6 | 26.3 | 24.3 | 24.1 | 25.0 | 27.3 | 24.4 | 22.7 | 25.0 | 25.3 | 26.6 | 26.4 | 26.0 | 25.4 |
| PPT (Ma et al., 2022) | Full | 23.2 | 26.3 | 22.0 | 22.9 | 25.2 | 23.1 | 23.8 | 28.5 | 31.2 | 25.2 | 27.4 | 23.6 | 26.5 | 23.4 | 25.0 | 25.2 |
| PPT + MCC (Ma et al., 2022; Su et al., 2021) | Interp | 23.5 | 27.5 | 21.5 | 21.9 | 24.7 | 29.0 | 23.0 | 23.7 | 30.6 | 34.1 | 26.6 | 22.2 | 22.4 | 27.6 | 24.0 | 25.5 |
| PPT + SLERP (Ma et al., 2022; Chen et al., 2022) | Interp | 23.0 | 26.0 | 21.6 | **21.6** | 24.9 | 27.1 | 22.8 | 23.3 | 28.2 | 32.4 | 24.9 | 22.1 | 23.1 | 26.2 | 24.7 | 24.8 |
| PPT | Sparse | 24.4 | 27.1 | 22.8 | 24.6 | 25.8 | 24.2 | 25.8 | 28.2 | 31.1 | 25.8 | 28.2 | 25.2 | 28.4 | 27.4 | 28.2 | 26.4 |
| Ours | Sparse | 23.2 | 22.5 | 21.0 | 21.9 | 20.5 | 21.2 | 20.5 | 22.0 | 21.2 | 19.7 | 21.4 | 20.5 | 22.1 | 21.8 | 20.7 | 21.3 |
| *2D—CPN* | | | | | | | | | | | | | | | | | |
| GLA-GCN (T=243) | Single | 41.4 | 44.4 | 40.8 | 41.8 | 46.0 | 54.1 | 42.1 | 41.5 | 57.9 | 62.9 | 45.1 | 42.8 | 29.3 | 45.9 | 29.9 | 44.4 |
| KTP-Former (T=243) | Single | 37.7 | 39.7 | 35.9 | 37.7 | 42.1 | 48.0 | 38.7 | 39.2 | 52.5 | 56.2 | 41.3 | 40.0 | 26.8 | 39.6 | 27.6 | 40.2 |
| FinePose (T=243) (Xu et al., 2024a) | Single | - | - | - | - | - | - | - | - | - | - | - | - | - | - | - | 40.2 |
| Adafuse | Full | 35.0 | 37.1 | 32.2 | 34.9 | 35.2 | 36.6 | 33.0 | 34.6 | 40.5 | 41.3 | 37.2 | 35.3 | 33.8 | 37.4 | 33.1 | 35.8 |
| Adafuse + MCC | Interp | **31.9** | 36.9 | 30.1 | **32.8** | 33.4 | **32.0** | 32.0 | 32.4 | **37.4** | 48.8 | **33.9** | 33.7 | 32.4 | 36.5 | 31.6 | 34.4 |
| Adafuse + SLERP | Interp | 34.4 | 36.8 | 31.9 | 34.0 | 34.6 | 35.7 | 32.3 | 34.1 | 40.2 | 41.0 | 34.8 | 33.4 | 34.8 | 36.8 | 32.4 | 35.3 |
| Adafuse | Sparse | 35.9 | 37.2 | 33.3 | 36.2 | 36.7 | 37.2 | 33.1 | 36.9 | 41.8 | 41.0 | 37.6 | 35.8 | 37.0 | 38.2 | 35.1 | 36.9 |
| Sgraformer (Zhang et al., 2024) | Full | - | - | - | - | - | - | - | - | - | - | - | - | - | - | - | 35.4 |
| Ours | Sparse | 32.0 | 35.2 | 30.0 | 32.4 | 33.0 | 34.4 | 30.2 | 32.3 | 38.9 | 40.1 | 35.5 | 32.9 | 31.4 | 36.0 | 30.4 | 33.6 |
| *2D—SimpleBaseline* | | | | | | | | | | | | | | | | | |
| GLA-GCN (T=243) | Single | 41.1 | 42.9 | 40.5 | 39.3 | 44.2 | 52.8 | 42.5 | 40.9 | 54.1 | 60.6 | 44.5 | 40.4 | 32.2 | 44.8 | 35.2 | 43.7 |
| KTP-Former (T=243) | Single | 35.4 | 36.9 | 34.3 | 34.7 | 39.6 | 43.4 | 36.3 | 35.0 | 47.5 | 57.4 | 39.4 | 35.4 | 27.5 | 38.7 | 29.4 | 38.1 |
| FinePose (T=243) | Single | 31.7 | 32.3 | 28.7 | 29.7 | 33.6 | 34.9 | 29.1 | 28.7 | 40.9 | 40.9 | 32.9 | 31.4 | 23.1 | 31.6 | 22.4 | 31.4 |
| Adafuse | Full | 28.3 | 29.9 | 25.3 | 29.5 | 26.9 | 26.4 | 27.0 | 28.1 | 28.7 | 32.1 | 27.8 | 28.8 | 26.7 | 29.6 | 25.5 | 28.1 |
| Adafuse + MCC | Interp | 27.6 | 30.0 | 25.1 | 29.4 | 26.5 | 26.9 | 26.4 | 27.9 | 29.2 | 32.3 | 27.9 | 28.5 | 26.4 | 29.9 | 25.4 | 28.0 |
| Adafuse + SLERP | Interp | 28.3 | 30.0 | 25.3 | 30.5 | 26.9 | 26.4 | 26.9 | 28.1 | 28.7 | 32.1 | 27.7 | 28.8 | 26.8 | 29.6 | 25.5 | 28.1 |
| Adafuse | Sparse | 29.9 | 30.6 | 26.1 | 31.2 | 27.9 | 27.4 | 27.7 | 30.2 | 29.7 | 32.3 | 28.5 | 29.8 | 30.8 | 31.0 | 29.0 | 29.5 |
| Algebraic (Iskakov et al., 2019a) | Full | 19.8 | 23.0 | 20.3 | 49.9 | 21.9 | 21.7 | **18.3** | 20.5 | 23.4 | 58.6 | 22.1 | 48.4 | 23.0 | 22.5 | 24.1 | 27.5 |
| Volumetric (Iskakov et al., 2019a) | Full | **18.8** | **21.7** | **19.6** | 50.1 | 21.2 | **21.0** | 18.4 | **20.2** | 21.8 | 57.1 | **21.5** | 48.4 | 22.5 | **21.7** | 22.5 | 26.7 |
| Sgraformer | Full | 24.3 | 25.1 | 21.1 | 24.7 | 24.5 | 24.9 | 21.6 | 22.1 | 26.5 | 32.1 | 25.1 | 24.0 | **21.3** | 25.4 | 21.6 | 24.3 |
| Ours | Sparse | 21.2 | 24.7 | 19.7 | **23.0** | **19.8** | 21.6 | 19.0 | 21.6 | 22.9 | **31.2** | 21.6 | **23.2** | 21.7 | 23.4 | **19.8** | **22.3** |

Note: Complete version with all baseline comparisons. Gray rows highlight our method. Action abbreviations: Directions (Dir), Discussion (Disc), Sitting Down (SitD), Walking Dog (WalkD), Walking Together (WalkT). Time frames (T=243) are shown where applicable. For the 2D pose estimation, we utilize ground truth, CPN (Cascaded Pyramid Network), and SimpleBaseline to obtain the corresponding 2D pose sequences. $T$ represents the number of input time frames. MCC (Motion Consistency and Continuity) and SLERP (Spherical Linear Interpolation) are keypoint interpolation methods. MCC is a neural network-based interpolation method, while SLERP is a traditional interpolation technique.

**Table 2:** P-MPJPE Comparison on with state-of-art pose estimation methods on Human3.6M (mm) using the ground-truth and detected 2D poses. Input types: Single-view (Single), Multi-view full-frame (Full), Multi-view interpolated (Interp). Best in bold.

| METHOD | INPUT | ACTIONS | | | | | | | | | | | | | | | AVG |
|---|---|---|---|---|---|---|---|---|---|---|---|---|---|---|---|---|---|
| | | Dir. | Disc. | Eat. | Greet. | Phone. | Photo. | Pose. | Pur. | Sit. | SitD. | Smoke. | Wait. | Walk. | WalkD. | WalkT. | |
| *2D—SimpleBaseline-P-MPJPE* | | | | | | | | | | | | | | | | | |
| GLA-GCN (T=243) (Yu et al., 2023b) | Single | 32.1 | 35.1 | 33.2 | 32.0 | 35.4 | 40.9 | 33.1 | 33.4 | 43.5 | 50.0 | 36.5 | 32.5 | 25.5 | 37.1 | 27.1 | 35.2 |
| KTP-Former (T=243) (Peng et al., 2024) | Single | 28.6 | 31.1 | 28.3 | 28.9 | 32.7 | 34.9 | 29.0 | 29.2 | 39.9 | 47.3 | 33.5 | 29.0 | 22.4 | 32.5 | 23.9 | 31.4 |
| FinePose (T=243) (Xu et al., 2024a) | Single | 24.8 | 26.3 | 24.7 | 24.0 | 27.0 | 28.1 | 22.4 | 23.8 | 33.7 | 33.3 | 27.4 | 24.6 | 19.2 | 25.8 | 18.5 | 25.6 |
| Adafuse (Zhang et al., 2021) | Full | 21.0 | 22.4 | 19.1 | 20.9 | 20.8 | 20.2 | 19.4 | 20.0 | 22.2 | 23.1 | 21.3 | 20.1 | 19.4 | 22.1 | 18.1 | 20.7 |
| Adafuse + MCC (Zhang et al., 2021; Su et al., 2021) | Interp | 20.5 | 22.6 | 18.8 | 21.0 | 20.5 | 21.0 | 19.1 | 19.4 | 22.6 | 23.3 | 21.6 | 20.1 | 19.4 | 22.5 | 18.4 | 20.7 |
| Adafuse + SLERP (Zhang et al., 2021; Chen et al., 2022) | Interp | 20.9 | 22.4 | 19.0 | 22.2 | 20.8 | 20.2 | 19.3 | 19.9 | 22.2 | **23.0** | 21.2 | 20.1 | 19.3 | 22.2 | 18.1 | 20.7 |
| Adafuse (Zhang et al., 2021) | Sparse | 22.5 | 22.6 | 19.7 | 22.5 | 21.5 | 20.9 | 20.0 | 20.5 | 22.5 | 23.4 | 21.7 | 20.6 | 23.3 | 23.4 | 20.9 | 21.7 |
| Sgraformer (Zhang et al., 2024) | Full | 19.9 | 20.1 | 18.1 | **18.2** | 20.8 | 20.3 | 17.1 | 17.7 | 23.0 | 26.2 | 21.9 | 18.6 | **17.7** | 20.7 | 17.9 | 19.9 |
| Ours | Sparse | 20.3 | 22.9 | 17.8 | **18.2** | 17.9 | 19.0 | 15.6 | 17.4 | 21.3 | 27.3 | 18.9 | 18.2 | 19.0 | 20.5 | 16.6 | **19.4** |

where $\hat{\mathbf{H}}_u^j(x)$ represents the corrected heatmap at location $x$ in view $u$, $\lambda$ is a balancing parameter between the current and other views, and $\mathbf{p}^u(x)$ denotes the epipolar line of $x$ in view $u$. $M$ is the number of camera views. The term $\max_{x' \in \mathbf{p}^u(x)} \mathbf{H}_u^n(x')$ represents the maximum response along the epipolar line in view $u$.

Following spatial correction, this process ensures that the expanded set $\mathbf{H}$ of heatmaps accurately captures spatial information across multiple views. By compensating for missing frames, this approach enhances the model's robustness and improves the quality of 3D pose estimation in sparse, interleaved multi-view scenarios. Consequently, it enables the generation of a spatially rectified, dense, full-frame heatmap input derived from the initial sparse, interleaved heatmaps.

## 3.2 WARPER: TEMPORAL POSE AGGREGATION

After performing epipolar geometry-based heatmap fusion as described in Section 3.1.2, we obtain a set of partly corrected heatmaps $\hat{\mathbf{H}}_{V_j}^i(x)$ that integrate spatial information across multiple views

$\mathbf{V} = \{V_1, V_2, \ldots, V_M\}$. We introduce a Warper module designed to capture temporal pose information from adjacent frames, further enhancing the temporal consistency of the sparse heatmaps (Pöppel, 1994; Zhang et al., 2019).

For each view $V_j$ at a target frame $n$, when $n \neq M \cdot i + j, n \in [M \cdot i, M \cdot i + (M-1)]$, we compute the difference between the corrected heatmap $\hat{\mathbf{H}}_{V_j}^n(x)$ and the heatmap from the sparse interleaved input $\mathbf{H}_{V_j}^{M \cdot i + j}(x)$ to capture temporal changes between frames, which can be denoted as:

$$\Phi_{V_j}^n(x) = \hat{\mathbf{H}}_{V_j}^n(x) - \mathbf{H}_{V_j}^{M \cdot i + j}. \tag{9}$$

This difference is then passed through a series of $3 \times 3$ residual blocks to extract temporal features. Following the residual blocks, we apply five $3 \times 3$ convolutional layers (Yu et al., 2017) with dilation rates $d \in \{3, 6, 12, 18, 24\}$, allowing the model to predict a set of offset maps $\{o_{V_j}^{(d)}(x)\}_{d=1}^5$ for each pixel $x$ in the heatmap. These offsets are used to warp the target heatmap deformably (Dai et al., 2017), aligning it with temporal features. The details of the Warper module are shown in Figure 4. Thus, for each spatially corrected heatmap $\mathbf{H}_{V_j}^n$, we generate five temporally aligned warped heatmaps by applying the offsets. These warped heatmaps are then aggregated by summation:

$$\tilde{\mathbf{H}}_{V_j}^n = \sum_{d=1}^5 \mathbf{Warper}(\Phi_{V_j}^n, o_{V_j}^{(d)}(x)), \tag{10}$$

where $\mathbf{Warper}(\cdot, o_{V_j}^{(d)}(x))$ represents the warping operation that uses the offset $o_{V_j}^{(d)}(x)$ to align each pixel $x$ in the heatmap. Notably, we trained a distinct warper model for each temporal mode.

This warping and aggregation process is applied to all interleaved frames in the sparse input set, ensuring that each refined heatmap $\tilde{\mathbf{H}}_{V_j}^n$ integrates both spatial and temporal information. The resulting tensor captures comprehensive pose information across views and time frames, improving the robustness and accuracy of 3D pose estimation under sparse interleaved multi-view settings.

## 4 EXPERIMENT

### 4.1 DATASET

**Human3.6M.** Human3.6M is a large-scale benchmark dataset widely used for 3D human pose estimation in controlled indoor environments. It consists of 3.6 million frames recorded from four synchronized high-resolution cameras capturing 11 professional actors (6 male, 5 female) performing 15 distinct activities, including walking, sitting, and object interactions. This dataset includes accurate 3D skeletal annotations obtained via a motion capture system and full-body 3D scans of each actor, enabling precise analysis of both 2D and 3D poses.

**MPI-INF-3DHP.** The MPI-INF-3DHP dataset is a comprehensive resource for multi-view 3D human pose estimation, featuring annotated frames from indoor and everyday settings. It includes 8 actors (4 male, 4 female), each performing 8 activity sets, such as walking, sitting, complex exercises, and dynamic actions. With diverse scenarios and multi-view recordings, the dataset enables robust evaluation of models under varying conditions.

**Table 3:** Reconstruction Error (MPJPE in mm) on the MPI-INF-3DHP Dataset. Input 2D pose sequences are obtained using a SimpleBaseline detector. $T$ denotes the number of input frames. Best results are highlighted in bold.

| MPJPE (SimpleBaseline(2D)) | Input Method | MPJPE ↓ |
|---|---|---|
| GLA-GCN (T=243) | Single | 75.00 |
| KTP-Former (T=243) | Single | 67.59 |
| Adafuse | Full | 78.57 |
| Adafuse + MCC | Interpolation | - |
| Adafuse + SLERP | Interpolation | 83.37 |
| PPT | Full | 106.30 |
| PPT + MCC | Interpolation | - |
| PPT + SLERP | Interpolation | 110.34 |
| Ours | sparse Interleaved | **65.89** |

It is worth noting that the MPI-INF-3DHP dataset has not been extensively trained with a 2D detector. We are **the first to process and align this dataset**, and we will open-source all our models and codes in the experiments.

**Table 4:** Model Parameter Count and Performance Efficiency. Performance Efficiency (MPJPE/mm per MB) is calculated as the ratio of MPJPE (in mm) to model size (in MB). Smaller values of this metric indicate better trade-offs between performance (MPJPE) and model size (MB), with more efficient models achieving lower MPJPE while maintaining smaller parameter sizes. The average latency specifically refers to the computational time of a single model inference (in milliseconds).

| Method | Para.(M) ↓ | Flops.(GFLOPs)↓ | Average Latency. (ms)↓ | Performance per MB (MPJPE/mm per MB) ↓ |
|---|---|---|---|---|
| GLA-GCN (T=243) | 69.99 | **51.13** | **24.10** | 0.624 |
| KTP-Former (T=243) | 103.85 | 51.64 | 24.11 | 0.367 |
| FinePose (T=243) | 269.23 | 287.32 | 82.24 | **0.117** |
| Adafuse (T=1) | **69.66** | 204.26 | 96.028 | 0.403 |
| Adafuse + SLERP | **69.66** | 204.26 | 96.03 | 0.403 |
| Adafuse + MCC | 72.25 | 204.26 | 96.028 | 0.388 |
| Sgraformer + Full | 81.23 | 204.28 | 99.19 | 0.299 |
| Ours | 76.51 | **111 .36** | **44.51** | **0.291** |

## 4.2 EVALUATION METRICS.

The **MPJPE** measures 3D pose accuracy via mean Euclidean distance between predicted ($\hat{P} = \{\hat{p}_1, \ldots, \hat{p}_J\}$) and ground truth ($P = \{p_1, \ldots, p_J\}$) joints:

$$\text{MPJPE} = \frac{1}{J} \sum_{i=1}^{J} \|\hat{p}_i - p_i\|_2 \tag{11}$$

where $\| \cdot \|_2$ is Euclidean norm.

## 4.3 EXPERIMENTAL RESULTS

**Results on the Human3.6M Dataset.** We conducted extensive experiments on the Human3.6M dataset using different 2D pose detectors to evaluate the effectiveness of our sparse interleaved approach. As shown in Table 1, our method consistently achieves new state-of-the-art performance across various settings.

First, with ground truth (GT) 2D poses, our model achieves a minimum average MPJPE of 21.3mm. This result is not only significantly better than all single-view methods (e.g., an improvement of **25.2**% over GLA-GCN's 28.5mm) but also outperforms multi-view approaches like Adafuse (23.7mm), yielding a performance gain of approximately 10.1%. Notably, our method achieves the best performance on 12 out of 15 action categories, with a particularly low MPJPE of 19.7**mm** on the challenging "SitDown" action, which is notably better than Adafuse's 21.6mm.

Second, we used CPN and SimpleBaseline as 2D pose detectors to validate the model's robustness in real-world scenarios. With CPN, our method still achieves the best average MPJPE of 33.6mm, representing a substantial **24.3**% performance improvement over the single-view GLA-GCN (44.4mm). Similarly, with SimpleBaseline, we achieve an impressive 22.3mm MPJPE, a significant improvement of approximately 20.6% over Adafuse (28.1mm). Furthermore, our method achieves the best result in 14 out of 15 action categories with this detector.

We also evaluated our model using the P-MPJPE metric, as shown in Table 2, which assesses the accuracy of relative joint positions. With SimpleBaseline 2D inputs, our model's average P-MPJPE is only 19.4mm, representing a 2.5% and 6.3% improvement over Sgraformer (19.9mm) and Adafuse (20.7mm), respectively. Our method achieves the best P-MPJPE in 13 out of 15 action categories, for instance, a remarkable 15.6mm on the "Pose" action, which is significantly better than Sgraformer's 17.1mm. This result provides strong evidence that our sparse interleaved approach not only precisely reconstructs the absolute 3D pose but also accurately captures the intricate internal structure and relative relationships of the human joints.

**Results on MPI-INF-3DHP.** As shown in Table 3. Our method demonstrates generalization capabilities on the more challenging MPI-INF-3DHP dataset, which features diverse outdoor settings and complex motions. With SimpleBaseline, we achieve 65.89mm MPJPE, substantially outperforming both single-view methods (GLA-GCN: 75.00mm, KTP-Former: 67.59mm) and multi-view approaches (Adafuse: 78.57mm, PPT: 106.30mm). This advantage extends to comparisons with interpolation-based methods (Adafuse+SLERP: 83.37mm, PPT+SLERP: 110.34mm), highlighting the effectiveness of our sparse interleaved input strategy in challenging real-world scenarios.

**Model Efficiency Analysis.** As shown in Table 4, our method exhibits significant advantages across multiple performance indicators. First, with a parameter count of only 76.51M, it achieves an effective balance between model capacity and performance, being considerably more lightweight

and efficient than (FinePose) (269.23M). This demonstrates that our model attains high accuracy with substantially fewer parameters. In terms of performance efficiency (MPJPE/mm per MB), our method achieves an impressive value of 0.291, notably surpassing (GLA-GCN) (0.624) and (KTP-Former) (0.367). This highlights the model's ability to provide higher accuracy per unit of model capacity. Although (FinePose) attains a slightly lower MPJPE, its excessive parameter count and computational demand result in less favorable efficiency. Furthermore, our method achieves the lowest average latency of 44.51 ms, which is significantly faster than (Adafuse) (96.03 ms) and (FinePose) (82.24 ms), ensuring real-time responsiveness. In addition, our approach reaches a processing speed four times that of the input FPS ($4f$), further confirming its potential for real-time pose estimation applications. Overall, our method is highly suitable for scenarios that require both high performance and low latency.

**Ablation Analysis.** To validate the effectiveness of our key components, we conduct ablation studies on both datasets using SimpleBaseline as the 2D detector. On Human3.6M, starting from a baseline MPJPE of 36.06mm, the addition of spatial heatmap fusion improves performance to 31.54mm, demonstrating the effectiveness of our epipolar geometry-based fusion strategy. Further, incorporating the Warper module reduces the MPJPE to 22.28 mm, yielding a 38.2% improvement compared to the Spatial Fusion only baseline. The detailed results are presented in Table 5.

On the MPI-INF-3DHP dataset, our spatial fusion module reduces the error from 94.46mm to 88.63mm, while the complete model achieves a significantly lower error of 65.89mm, yielding an overall improvement of 30.25%. These quantitative results substantiate the efficacy of both our temporal and spatial fusion strategies across diverse datasets. The detailed experimental setup and specifics are provided in the Appendix.

# 5 CONCLUSION

In this paper, we addressed a fundamental limitation of conventional 3D human pose estimation: the reliance on dense, synchronized multi-view inputs. To this end, we pioneered a sparse interleaved input paradigm that leverages the rich spatio-temporal dependencies often overlooked by traditional

**Table 5:** Ablation study results. We conducted ablation studies on the Human3.6M and MPI-INF-3DHP datasets to validate the effectiveness of the proposed space fusion module based on epipolar geometry and the temporal fusion module Warper. We use SimpleBaseline as 2D baseline model. We have bolded the best results.

| Method | Spatial Heatmap Fusion | Warper | Avg. ↓ |
|---|---|---|---|
| | ✗ | ✗ | 36.06 |
| Ours (Human3.6M) | ✓ | ✗ | 31.54 |
| | ✓ | ✓ | **22.28** |
| | ✗ | ✗ | 94.46 |
| Ours (MPI-INF-3DHP) | ✓ | ✗ | 88.63 |
| | ✓ | ✓ | **65.89** |

methods. Our work makes two key contributions to the field. First, we introduced a new task formulation by demonstrating that a high-frequency pose output can be accurately reconstructed from sparse, temporally-interleaved inputs. Second, we designed the DenseWarper module, an efficient end-to-end framework capable of transforming these sparse inputs into dense outputs with high spatio-temporal coherence. Our extensive experiments on the Human3.6M and MPI-INF-3DHP datasets rigorously validate that our approach surpasses conventional dense input methods and achieves state-of-the-art performance. This breakthrough highlights our method's ability to challenge the long-held assumption of dense synchronized inputs fundamentally. Ultimately, our research provides a compelling proof-of-concept for the future of resource-efficient, real-time 3D perception, paving the way for applications in domains requiring high temporal resolution, such as robotics and VR/AR.

**Limitation.** Our method has not been explored under non-uniform intervals or extremely low-frame-rate camera sampling. For cases with extensive inter-camera sampling intervals, as shown in Figure 5 in the appendix, our method may struggle to extract and recover spatio-temporal information effectively. Consequently, our approach is, to a certain extent, dependent on the density of the multi-view input stream, particularly in the temporal dimension.

**Future Work.** Our work opens up several promising directions for future research. First, we plan to extend our sparse interleaved input paradigm to other multi-view 3D tasks, such as object detection, to explore its generalizability and provide new insights into these fields. Second, we will delve deeper into the theoretical underpinnings of our novel input method, examining it from the perspective of interpretability. Future work will also focus on enhancing the model's robustness under more challenging conditions, including sparse and irregular sampling.

ACKNOWLEDGMENTS

This work was supported in part by Beijing Municipal Science Technology Commission No. Z251100004625090, by the National Science Foundation of China (NSFC) under Grant No. 62176134, and by a grant from the Assisted Medical Consultation Project Based on DeepSeek.

ETHICS STATEMENT

Our research aims to advance 3D human pose estimation by proposing a sparse interleaved input paradigm that addresses the insufficient spatio-temporal information utilization and frame rate limitations of traditional methods. All data used in this paper is from publicly available benchmark datasets (Human3.6M and MPI-INF-3DHP) where personal identifying information was anonymized prior to release. We made no modifications to these datasets to re-identify any individual. We confirm that this study did not involve the direct participation of human subjects. We believe our proposed technology, which includes the sparse interleaved input paradigm and the DenseWarper module, is designed solely to enhance computational efficiency and poses no risk of harmful applications, bias, or privacy infringement. We declare that the research and writing of this paper have adhered to the principles of academic integrity, with all citations and references clearly acknowledged.

REPRODUCIBILITY STATEMENT

To ensure the full reproducibility of our work, we provide all necessary information within this paper and its supplementary materials. The source code for our proposed DenseWarper framework, including the model implementation, training configurations, and evaluation scripts, will be made available via an anonymous link in the supplementary materials. Our experiments are based on the publicly available Human3.6M and MPI-INF-3DHP datasets, with all data processing steps also detailed in the supplementary materials. These measures ensure our work can be fully verified and built upon by the research community.

THE USE OF LARGE LANGUAGE MODELS (LLMS)

This paper utilized a large language model (LLM) as a general-purpose assist tool to aid or polish the writing. The LLM's role was strictly limited to improving the clarity, grammar, and style of the text. It was not used for research ideation, data analysis, or the generation of core scientific content. The authors take full responsibility for the content, originality, and accuracy of the paper.

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

# A    RELATED WORK

## A.1    3D HUMAN POSE ESTIMATION

Early approaches to 3D human pose estimation relied heavily on direct regression from 2D images to 3D coordinates (Li & Chan, 2014; 2015; Moon et al., 2019; Park et al., 2016; Pavlakos et al., 2018a; Tekin et al., 2016; Wehrbein et al., 2021; Luvizon et al., 2018; 2019; Moon & Lee, 2020; Pavlakos, Georgios and Zhou, Xiaowei and Derpanis, Konstantinos G. and Daniilidis, Kostas, 2017; Shen & Tang, 2024; Rommel et al., 2024). With the advent of deep learning, two-stage methods became prevalent, first detecting 2D keypoints and then lifting them to 3D space (Martinez et al., 2017; Chen et al., 2021; Hossain & Little, 2018; Liu et al., 2020b; Pavllo et al., 2019a; Zheng et al., 2021b; Li et al., 2022; Zhang et al., 2022; Tang et al., 2023; Shan et al., 2023; Cai et al., 2019; Hu et al., 2021; Liu et al., 2020a; Xu & Takano, 2021; Yu et al., 2023a; Zhao et al., 2019a; Zou & Tang, 2021; Gong et al., 2023; Li et al., 2023; Zhao et al., 2022; Zhu et al., 2021). Recent methods have incorporated additional constraints such as bone length consistency (Zhao et al., 2019b) and anatomical priors (Pavlakos et al., 2018b) to improve estimation accuracy. Furthermore, self-attention mechanisms (Zheng et al., 2020) and graph neural networks (Zou et al., 2020; Zhao & Tulsiani, 2024) have been introduced to capture long-range dependencies and structural relationships in human poses.

## A.2    MULTI-VIEW 3D POSE ESTIMATION

Multi-view approaches have demonstrated superior performance to single-view methods due to their ability to resolve depth ambiguity. Traditional methods typically use triangulation-based techniques (Hartley & Zisserman, 2003; Qiu et al., 2019a; Zhang et al., 2021; Mitra et al., 2020; Shuai et al., 2022; Zhou et al., 2023) to reconstruct 3D poses from synchronized multi-view 2D detections. Recent learning-based approaches have explored various ways to fuse multi-view information. Qiu et al. (Qiu et al., 2019b) proposed cross-view fusion using epipolar geometry, while Remelli et al. (Remelli et al., 2020a) introduced a lightweight architecture for real-time multi-view pose estimation. Rhodin et al. (Rhodin et al., 2018) leveraged geometric consistency across views to improve unsupervised learning of 3D pose estimation. However, these methods generally rely on dense, synchronized multi-view inputs, which can be computationally expensive and may not fully utilize temporal information.

## A.3    TEMPORAL INFORMATION IN POSE ESTIMATION

Temporal information is crucial for robust pose estimation, particularly in challenging scenarios with occlusions or motion blur. Previous work has explored various approaches to incorporate temporal information, including recurrent neural networks (Hossain & Little, 2018) and temporal convolutions (Pavllo et al., 2019b). Recent works like Zheng et al. (Zheng et al., 2021a) have proposed combining spatial and temporal attention mechanisms to capture motion dynamics better. However, most existing methods process temporal information within a single view, potentially missing valuable cross-view temporal correlations. Approaches like (Sun et al., 2018) have attempted to bridge this gap by incorporating temporal consistency constraints in multi-view settings.

## A.4    SPARSE AND EFFICIENT VISION METHODS

Recent trends in computer vision have shown increasing interest in efficient processing methods. Sparse convolutions (Graham et al., 2018) and attention mechanisms (Zhu et al., 2020) have been proposed to reduce computational overhead while maintaining performance. In multi-view tasks, sparse view selection (Zhang et al., 2020; Zhao & Tulsiani, 2024) and adaptive sampling strategies have been explored, though primarily for static scene reconstruction rather than dynamic human pose estimation. Notable works like (Jiang et al., 2023) have introduced probabilistic frameworks for efficient multi-view processing.

## A.5    GEOMETRIC FEATURE WARPING

Our work builds upon recent advances in geometric feature warping, particularly in the context of multi-view feature fusion. Epipolar geometry has been extensively used for cross-view feature

alignment (He et al., 2020), while deformable convolutions (Dai et al., 2017) have shown promise in handling dynamic spatial transformations. Recent works (Chen et al., 2020) have explored the integration of geometric constraints with learning-based feature warping. However, combining these techniques for spatio-temporal feature warping in sparse multi-view scenarios remains largely unexplored.

# B  CODES AND MODELS

We have organized the code for our method in the *codes* directory and compiled the checkpoint files used in the experiments, as shown in Table 6. Upon publication of the paper, we will release the code and pretrained models. Detailed usage instructions for the code are provided in the "*/codes/DenseWarper/README.md*" file for reference.

| Model Name | Dataset | Performance (Metric) | Checkpoint Name | Model Source |
|---|---|---|---|---|
| GLA-GCN(T=243) | MPI-INF-3DHP | MPJPE: 75.00 mm | GLA-GCN_3DHP.bin | Reproduced |
| KTP-Former(T=243) | MPI-INF-3DHP | MPJPE: 67.59 mm | KTP-Former_3DHP.bin | Reproduced |
| Adafuse | MPI-INF-3DHP | MPJPE: 78.57 mm | Adafuse_3DHP.pth.tar | Reproduced |
| Adafuse (Zhang et al., 2021) + MCC (Su et al., 2021) | MPI-INF-3DHP | MPJPE: - mm | MCC_3DHP.pth.tar | Reproduced |
| Adafuse + SLERP (Chen et al., 2022) | MPI-INF-3DHP | MPJPE: 83.37 mm | - | Reproduced |
| PPT | MPI-INF-3DHP | MPJPE: 106.30mm | PPT_3DHP.pth.tar | Reproduced |
| PPT + MCC | MPI-INF-3DHP | MPJPE: - mm | MCC_3DHP.pth.tar | Reproduced |
| PPT + SLERP | MPI-INF-3DHP | MPJPE: 110.34 mm | - | Reproduced |
| Ours | MPI-INF-3DHP | MPJPE: 65.89 mm | Ours_3DHP.pth.tar | Reproduced |
| GLA-GCN(T=243) (CPN) | Human3.6M | MPJPE: 40.39 mm | GLA-GCN_CPN.bin | Original |
| FinePose(T=243) (CPN) | Human3.6M | MPJPE: 40.20 mm | - | Reproduced |
| KTP-Former(T=243)(CPN) | Human3.6M | MPJPE: 40.18 mm | KTP-Former_CPN.bin | Original |
| Adafuse (CPN) | Human3.6M | MPJPE: 35.81 mm | Adafuse_H36M.tar | Original |
| Adafuse + MCC (CPN) | Human3.6M | MPJPE: 34.42 mm | MCC_H36M.pth | Original |
| Adafuse + SLERP (CPN) | Human3.6M | MPJPE: 35.27 mm | - | Original |
| Sgraformer (CPN) | Human3.6M | MPJPE: 35.40 mm | - | Reproduced |
| Ours (CPN) | Human3.6M | MPJPE: 33.57 mm | Ours_H36M.pth.tar | Reproduced |
| GLA-GCN(T=243) (SimpleBaseline) | Human3.6M | MPJPE: 43.74 mm | GLA-GCN.bin | Reproduced |
| FinePose(T=243) (SimpleBaseline) | Human3.6M | MPJPE: 31.40mm | - | Reproduced |
| KTP-Former(T=243)(SimpleBaseline) | Human3.6M | MPJPE: 38.08 mm | KTP-Former.bin | Reproduced |
| Adafuse (SimpleBaseline) | Human3.6M | MPJPE: 28.06 mm | Adafuse_H36M.pth.tar | Original |
| Adafuse + MCC (SimpleBaseline) | Human3.6M | MPJPE: 27.95 mm | MCC_H36M.pth | Original |
| Adafuse + SLERP (SimpleBaseline) | Human3.6M | MPJPE: 28.10 mm | - | Original |
| Sgraformer (SimpleBaseline) | Human3.6M | MPJPE: 24.32 mm | - | Reproduced |
| Ours (SimpleBaseline) | Human3.6M | MPJPE: 22.28 mm | Ours_H36M.pth.tar | Reproduced |

**Table 6:** Pretrained Models List. We conducted extensive experiments on the Human3.6M and MPI-INF-3DHP benchmark datases using a combination of open-source models and reproduced code. Here, "reproduced" means we reimplemented and retrained the model code, while "original" indicates we used open-source models for testing.

# C  EXPERIMENTAL DETAILS

## C.1  EXPERIMENTAL SETTINGS

In our experiments, we proposed the DenseWarper module for spatiotemporal fusion. All models were trained using two RTX A100 GPUs. Table 7 provides a comprehensive overview of the parameter settings used for training DenseWarper compared to other models.

| Method | Loss | $LR$ | $Epoch$ | $Batch$ | $Optimizer$ |
|---|---|---|---|---|---|
| GLA-GCN(T=243) | MPJPE | 0.01 | 200 | 512 | Ranger |
| KTP-Former(T=243) | WMPJPE+MPJVE+ temporal consistency loss | 0.00008 | 200 | 1024 | AdamW |
| Adafuse | MSE | 0.0001 | 50 | 4 | Adam |
| Adafuse + MCC | MSE | 0.0001 | 50 | 4 | Adam |
| Adafuse + SLERP | MSE | 0.0001 | 50 | 4 | Adam |
| PPT | 2dSmoothLoss | 0.001 | 200 | 32 | Adam |
| PPT + MCC | 2dSmoothLoss | 0.001 | 200 | 32 | Adam |
| PPT + SLERP | 2dSmoothLoss | 0.001 | 200 | 32 | Adam |
| Ours | 2dSmoothLoss | 0.001 | 50 | 4 | Adam |

**Table 7:** Experimental parameter settings for comparative analysis with different models.

## C.2 Other Experimental Details

**Experimental Details on Human3.6M (Ionescu et al., 2013).**

In this dataset, we employed three types of 2D pose detection methods to evaluate the performance of our models: GT, CPN (Chen et al., 2018), and SimpleBaseline (Xiao et al., 2018).

Among them, GT and CPN provide only the 2D coordinates of each keypoint, while SimpleBaseline directly outputs heatmaps for each keypoint. To ensure consistency in data formats, we generated heatmaps for the 2D coordinates of GT and CPN using a Gaussian distribution. Conversely, for SimpleBaseline, we extracted the 2D coordinates by identifying the locations of the maximum values in its heatmaps. This preprocessing ensures that GT, CPN, and SimpleBaseline data all have two forms: 2D coordinates and heatmaps.

Regarding model inputs, KTP-Former (Peng et al., 2024) and GLA-GCN (Yu et al., 2023b) require 2D coordinates, while Adafuse and DenseWarper use heatmaps as input. In contrast, the PPT model adopts an end-to-end architecture, taking only the raw images as input without relying on 2D pose detection results.

This setup allows a comprehensive evaluation of the models under different input formats.

**Experimental Details on MPI-INF-3DHP (Mehta et al., 2017).**

The processing method for this dataset is consistent with that of the Human3.6M dataset. We selected the same 17 keypoints corresponding to those in Human3.6M and used four camera views, specifically views 0, 2, 7, and 8.

For this dataset, we employed the SimpleBaseline method for 2D pose detection, using a model trained in-house. SimpleBaseline generates heatmaps for each keypoint. To facilitate further processing, we extracted the 2D coordinates by identifying the locations of the maximum values in the heatmaps, thereby obtaining both 2D coordinates and heatmaps as input formats.

In terms of model inputs, KTP-Former and GLA-GCN take 2D coordinates as input, while Adafuse and DenseWarper require heatmaps. The PPT (Ma et al., 2022) model, in contrast, employs an end-to-end architecture that uses only raw images as input without relying on any 2D pose detection results.

To ensure a fair comparison, all models were trained and tested using results derived from SimpleBaseline.

## D Epipolar Geometry

In the main text, we introduced the fundamental principles of epipolar geometry and the line-of-sight methods that employ epipolar geometry for multi-view fusion. Here, we expand upon these topics and provide a more detailed exposition.

### I. Mathematical Foundation of Epipolar Geometry

Epipolar geometry defines the geometric relationships between two camera views, essential for multi-view feature matching and reconstruction. Using the pinhole camera model, a 3D point $\mathbf{X} = [X, Y, Z, 1]^T$ is projected onto the image plane as a 2D point $\mathbf{q} = [u, v, 1]^T$:

$$\mathbf{q} = \mathbf{PX}, \tag{12}$$

where $\mathbf{P} = \mathbf{K}[\mathbf{R} \mid \mathbf{t}]$ is the $3 \times 4$ projection matrix. The intrinsic matrix $\mathbf{K}$ defines focal lengths and principal points, while the extrinsic parameters $[\mathbf{R} \mid \mathbf{t}]$ describe the camera's rotation and translation in the world coordinate system.

Given two cameras with projection matrices $\mathbf{P}$ and $\mathbf{P}'$, a 3D point $\mathbf{X}$ is projected onto the two images as $\mathbf{q}$ and $\mathbf{q}'$:

$$\mathbf{q} = \mathbf{PX}, \quad \mathbf{q}' = \mathbf{P}'\mathbf{X}. \tag{13}$$

Eliminating $\mathbf{X}$ leads to the epipolar constraint:

$$\mathbf{q}'^{\top}\mathbf{F}\mathbf{q} = 0, \tag{14}$$

where $\mathbf{F}$ is the $3 \times 3$ fundamental matrix, encapsulating the geometric relationship between uncalibrated cameras. If intrinsic parameters are known, the essential matrix $\mathbf{E}$ can be derived:

$$\mathbf{E} = [\mathbf{t}]_\times \mathbf{R}, \quad \mathbf{E} = \mathbf{K}'^\top \mathbf{F} \mathbf{K}. \tag{15}$$

Here, $[\mathbf{t}]_\times$ represents the skew-symmetric matrix of the translation vector $\mathbf{t}$.

The epipolar geometry also defines epipoles and epipolar lines. The epipoles, $\mathbf{e}$ and $\mathbf{e}'$, are the projections of one camera's optical center onto the other's image plane, satisfying $\mathbf{F}\mathbf{e} = \mathbf{0}$ and $\mathbf{F}^\top \mathbf{e}' = \mathbf{0}$. For a point $\mathbf{q}$ in the first image, the corresponding point $\mathbf{q}'$ in the second must lie on the epipolar line:

$$\mathbf{l}' = \mathbf{F}\mathbf{q}, \quad \mathbf{l} = \mathbf{F}^\top \mathbf{q}'. \tag{16}$$

This reduces the search space for matching points from 2D to 1D, significantly improving efficiency.

II. EPIPOLAR GEOMETRY FOR MULTI-VIEW FUSION

Epipolar geometry is crucial in multi-view fusion for feature matching, 3D reconstruction, and optimization:

**1. Feature Matching:** The epipolar constraint reduces the matching search space to epipolar lines, filtering mismatches by validating $\mathbf{q}'^\top \mathbf{F} \mathbf{q} = 0$.

**2. Triangulation:** Matched points are used to estimate the 3D position $\mathbf{X}$ via linear triangulation:

$$\begin{cases} \mathbf{q} \times (\mathbf{P}\mathbf{X}) = \mathbf{0} \\ \mathbf{q}' \times (\mathbf{P}'\mathbf{X}) = \mathbf{0}. \end{cases} \tag{17}$$

To refine accuracy, non-linear optimization minimizes the reprojection error:

$$\min_{\mathbf{X}} \sum_k \|\mathbf{q}_k - \pi_k(\mathbf{X})\|^2, \tag{18}$$

where $\pi_k(\mathbf{X})$ projects $\mathbf{X}$ onto the $k$-th camera.

**3. Optimization:** Incorporating the epipolar constraint into the optimization objective ensures geometric consistency:

$$\min_{\mathbf{X}} \sum_k \|\mathbf{q}_k - \pi_k(\mathbf{X})\|^2 + \lambda \sum_{i<j} \left(\mathbf{q}_j^\top \mathbf{F}_{ij} \mathbf{q}_i\right)^2. \tag{19}$$

This enhances robustness against noise and improves multi-view consistency.

Epipolar geometry provides the theoretical foundation for effective feature alignment, matching, and 3D reconstruction, enabling precise multi-view fusion in computer vision applications.

**4. Reprojection Distance.** In multi-view matching tasks, the lines $\mathbf{V}_1\mathbf{q}$ and $\mathbf{V}_2\mathbf{q}'$ may not intersect precisely at the 3D point $\mathbf{Q}$. To obtain the optimal estimate of $\mathbf{Q}$, we define the reprojection distance $d_{\text{Reproj}}$ as the minimum sum of squared distances from the estimated 3D human keypoint $\hat{\mathbf{Q}}$ to the projections $\mathbf{q}$ and $\mathbf{q}'$:

$$d_{\text{Reproj}}^2 = \min_{\hat{\mathbf{Q}}} \left( d^2(\mathbf{q}, \mathbf{P}\hat{\mathbf{Q}}) + d^2\left(\mathbf{q}', \mathbf{P}'\hat{\mathbf{Q}}\right) \right), \tag{20}$$

where $d(\cdot)$ denotes the Euclidean distance, and $d_{\text{Reproj}}$ represents the reprojection error between $\mathbf{q}$ and $\mathbf{q}'$.

**5. Sampson Distance.** To simplify the calculation, we adopt an approximation of the reprojection distance known as the Sampson distance, defined as:

$$d_{\text{Sampson}} = \frac{\mathbf{q}'^\top \mathbf{F} \mathbf{q}}{(\mathbf{F}\mathbf{q})_1^2 + (\mathbf{F}\mathbf{q})_2^2 + (\mathbf{F}^\top \mathbf{q}')_1^2 + (\mathbf{F}^\top \mathbf{q}')_2^2}, \tag{21}$$

where $\mathbf{F}$ is the fundamental matrix, and subscripts 1 and 2 refer to the first and second elements of a vector, respectively. Sampson distance allows us to measure the geometric error between two points without explicitly solving for the intermediate 3D point $\hat{\mathbf{Q}}$.

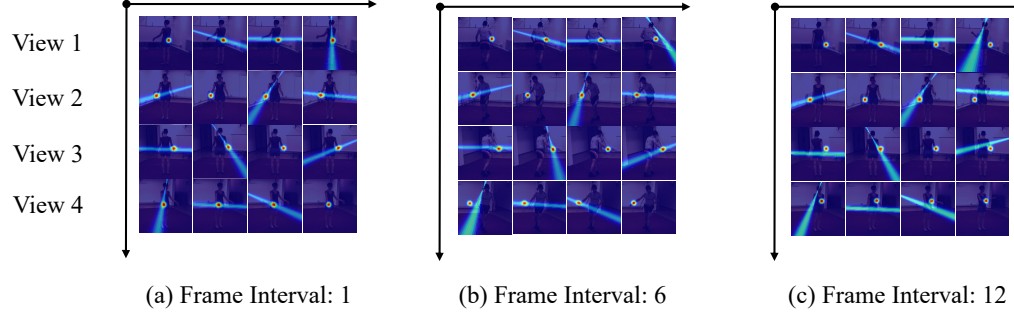

(a) Frame Interval: 1       (b) Frame Interval: 6       (c) Frame Interval: 12

**Figure 5:** Results of spatiotemporal heatmap fusion and correction using different frame intervals on the Human3.6M dataset. The camera sampling interval in the Human3.6M dataset is 50ms. Panels (a), (b), and (c) represent the results of spatial heatmap fusion with frame intervals of 1 frame, 6 frames, and 12 frames, respectively.

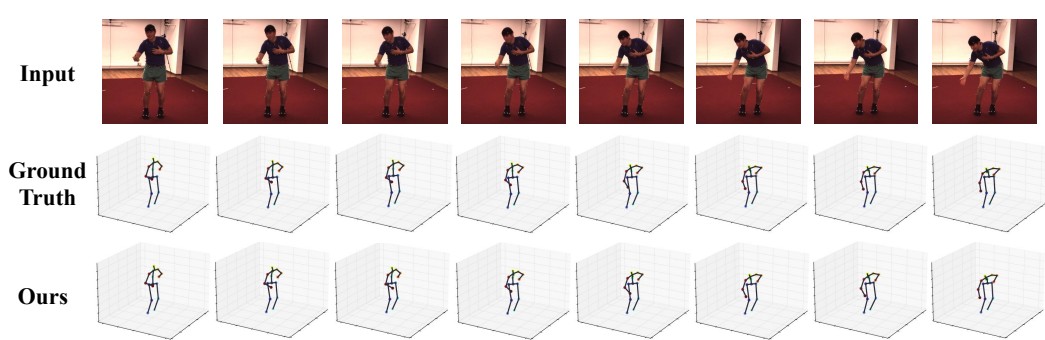

**Figure 6:** Visualization results showing the effects during continuous motion.

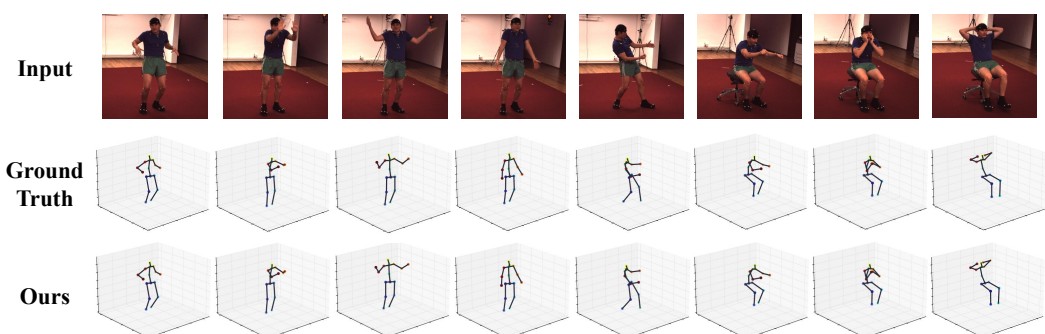

**Figure 7:** Visualization results demonstrating the effects during complex motion.

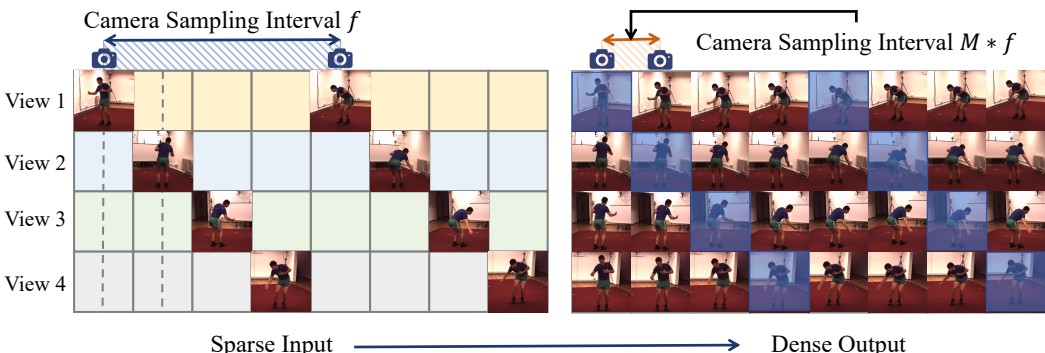

**Figure 8:** Illustration of frame rate enhancement through interleaved multi-view input. The camera frame rate $f = 1/\delta t$, where $\delta t$ represents the **camera sampling interval**. With a fixed camera frame rate, the input frame rate can be effectively increased using interleaved multi-view inputs, reaching up to $M \times f$, where $M$ denotes the number of camera viewpoints.

## E  SUPPLEMENTARY EXPERIMENTAL RESULTS

In the experiments, we further investigated the relationship between our epipolar geometry-based spatial heatmap fusion module and the temporal frames of the cameras. Theoretically, as the temporal frame interval increases (i.e., the camera sampling frequency decreases), the displacement of the heatmap points to be calibrated becomes larger, making heatmap fusion more challenging. As shown in Figure 5, the displacement of heatmaps from different views also increases under these conditions. In our experiments, the camera frame rate was set to 50 fps, which enabled effective heatmap fusion. This demonstrates that our method performs well under standard camera sampling conditions.

Additionally, we visualized the results of our method by comparing the 3D skeletons estimated from sparse interleaved inputs with the ground truth. The experimental results demonstrate that our method achieves accurate 3D skeleton estimation, effectively leveraging both the temporal and spatial information embedded in the sparse interleaved inputs. This validates the capability of our approach to transform sparse inputs into dense outputs. Figures 6 and 7 respectively present visualizations of 3D skeletons for continuous actions and complex, challenging actions.

### E.1  SPARSE INPUT FOR UPSAMPLING

As shown in Figure 8, when the staggered intervals of cameras with different viewpoints are controllable, we can achieve data upsampling on the coefficient input mode, and this method is applicable to all 3D multi-view tasks.

### E.2  QUANTITATIVE ANALYSIS OF DIFFERENT SAMPLING INTERVALS

To investigate how the sampling interval affects performance, we conducted a quantitative analysis, with the results presented in Table 8. It can be observed that with a moderate time interval (e.g., when Frame Interval = 6), the input still has some degree of density, and the interleaved viewpoints are able to provide effective spatial correction. As a result, the model performance does not degrade significantly. However, when the time interval is large (i.e., when the camera sampling rate is low), the data density decreases, leading to sparsity. In such cases, the sparse interleaved sampling paradigm becomes ineffective, and no effective correction is provided between different viewpoints. Furthermore, due to the large time interval, the performance of the temporal fusion module is also challenged. This experimental result suggests that our method is suitable for high frame rate, dense input scenarios, which aligns with the motivation of our paper.

### E.3 MODEL PERFORMANCE UNDER NON-UNIFORM INTERVAL SCENARIOS

To address the concern regarding potential non-uniform intervals between different viewpoints, we systematically sampled from the existing dataset to simulate input scenarios with irregular temporal spacing.

Unlike the original uniform interval input (with a window size of Window Size = 4), we no longer select input frames from a sequential window of size 4. Instead, we set the sampling window size x to be greater than 4 (i.e., $x > 4$) to ensure that all four viewpoints are selected, with no two viewpoints appearing at the same time frame. Specifically, we randomly and non-uniformly select four viewpoints from a sequence of a time window size of x. This successfully simulates the case where the time intervals between input viewpoints are non-uniform. We performed experiments with x = 6, 10, 12, and the experimental results are shown in Table 9.

The experimental results indicate that non-uniform intervals have a certain impact on model performance. As the maximum non-uniform interval increases, the model performance degrades more, which confirms our hypothesis that as the interval time increases, the temporal fusion module of the model will be challenged. This is due to the randomness caused by the non-uniform interval time paradigm. However, overall, even with larger maximum non-uniform intervals, our model still achieves relatively good results.

| Frame Interval | Frame Interval = 1 | Frame Interval = 6 | Frame Interval = 12 |
|---|---|---|---|
| Sampling Interval (ms) | 20 | 120 | 240 |
| Performance (MPJPE, mm) | 22.3 | 39.92 | 69.84 |

**Table 8:** Quantitative Analysis of Different Sampling Intervals.

| Window Size | 4 | 6 | 10 | 12 |
|---|---|---|---|---|
| Maximum Non-Uniform Interval (ms) | 20 (Uniform Interval) | 40 | 120 | 160 |
| Performance (MPJPE, mm) | 22.3 | 26.77 | 29.99 | 31.58 |

**Table 9:** Model Performance under Non-Uniform Interval Scenarios. In the original paper, the input interval is uniform and 20ms.

