# OpenReview forum: "From Sparse to Dense: Spatio-Temporal Fusion for Multi-View 3D Human Pose Estimation with DenseWarper"
_ICLR.cc/2026/Conference — ICLR 2026 Poster_

### Official Review · Reviewer_FTR4 · 2025-10-22

**Soundness:** 3
**Presentation:** 3
**Contribution:** 4
**Rating:** 6
**Confidence:** 3

**Summary:**

The authors propose a novel 3d human pose estimation framework which leverages images captured from different camera views at various time points to capture rich spatio-temporal information and effectively boost performance. Their approach theorectically increase the output post frame rate by N times with N cameras and enhance the temperal resolution of the production. In addition, using a spare subset of available frames, their method can reduce data redundancy while simultaneously achieve better performance.
They also introduce DenseWarper model which leverages epipolor geometry for efficient spatio-temporal heatmap exchange. Extensive evaluations using Human3.6M and MPI-INF-3DHP show that their method outperforms SOTA methods.
However, there are certain parts that need some clarifications and authors should discuss how their method can be extended to analyze videos involving multi-persons.

**Strengths:**

•	They are the first that propose 3d pose estimation task based on sparse interleaved multi-view input
•	They design DenseWarper to convert sparse interleaved inputs into dense pose outputs with high spatio-temporal consistency.
•	They conduct rigorous experiments to demonstrate that their proposed technique achieves better performance.

**Weaknesses:**

•	There are some parts in current writeup that need clarification When the Spatial Fusion module (Eq. 8) runs on the expanded set H at n_th frame,  it is fusing the real heatmap H_n at view1 with the replicated heatmap H'_n at view 2(which is actually the heatmap from n+∆). This means the module is still applying epipolar constraints to heatmaps of a person at two different frames (n and n+∆). The epipolar geometry is invalid for a moving object at different times. It seems like this paper implicitly assumes that this step can still refine approximate feature correspondences, even if they are not geometrically perfect.
•	Authors have only evaluated their method using datasets involving single person performing an action in each video. They should comment on how to extend their method to videos involving multiple persons. In this case, simply doing spatial fusion using their method will not work. There may be multiple heatmaps (one for each person) and if the people are close by, it may be hard to differentiate how to fuse nearby points correctly.

**Questions:**

•	Motivate why CPN is used as 2D detector. It seems to perform poorer than simplebaselines
•	Discuss how your method can be extended to handle multi-person videos.
* Perhaps run one set of experiments using multi-person activity dataset such as CMU Panoptic to find out how your method performs compared to existing Multiview 3D pose estimation methods such as MV-SSM: Multi-View State Space Modeling for 3D Human Pose Estimation, a CVPR 2025 paper.

---

> ### Author Response · Authors · 2025-11-13
> **Response to Reviewer FTR4**
>
> We sincerely thank you for your careful review and valuable feedback on our paper. Below, we address your specific questions and suggestions for improvement one by one. In the following content, “Q” denotes a *Question* and “W” denotes a *Weakness*.
>
>
>
> **Q1: Regarding the use of CPN as the 2D detector.**
>
>  In 3D human pose estimation (HPE) tasks, both CPN and SimpleBaseline have been used as 2D detectors in prior research. To ensure a fair comparison, we conducted experiments using both detectors in our work.
>
> **Q2, W2: On extending the method to multi-person scenarios.**
>
>  Thank you for this valuable suggestion. At present, our method—as well as most compared approaches—focuses on single-person scenarios. We agree that extending to multi-person scenes is both more challenging and meaningful. In such settings, occlusions occur frequently, and multi-view fusion can introduce additional noise. We plan to further explore how to effectively integrate multi-view information in multi-person settings in future work.
>
> **Q3: On experiments in multi-person scenes.**
>
> 1. We appreciate your suggestion and have carefully reviewed the MV-SSM paper. During our investigation, we identified several challenges: The method is not yet open-sourced, making it difficult to deploy and train within a short timeframe.
>
> 2. Our current framework is designed specifically for single-person scenarios and, therefore, not directly applicable to multi-person cases.
>
> Given these constraints, we were unable to conduct a direct comparison with MV-SSM at this stage. Nonetheless, we appreciate your suggestion and will continue to explore how our approach can be extended to multi-person scenarios — an idea with high potential for innovation and research value.
>
>
>
> **W1: On the invalidity of epipolar geometry across different time frames.**
>
>  We appreciate your insightful observation about the limitations of epipolar geometry for objects captured at different timestamps. You are correct — strict geometric triangulation is not valid when temporal discrepancies exist between views. However, it is important to note that in our approach, epipolar geometry is **not** used for strict triangulation. Instead, it serves as the first step of our method, providing *spatial correction guidance*.
>
> In the **Spatial Fusion** module, for instance, to obtain the spatially aligned heatmap $H_{1,2}$ at view $V_1$ at time $t_n + \delta$ (denoted as $H_{1,2}$), we project the heatmap from view $V_{2}$ at the same time ($H_{2,2}$) onto the image space of $H_{1,2}$ using an epipolar constraint. This limits cross-view information exchange to the most probable epipolar regions, acting as an effective spatial filter. It produces a noise-reduced and approximately aligned heatmap, though it inevitably carries temporal inaccuracies due to the time offset $\delta$.
>
> To handle this issue, we further introduce the **Temporal Warper** module, which refines and compensates for the temporal misalignment in the heatmaps, ensuring accurate final pose estimation.

---

> > ### Comment · Reviewer_FTR4 · 2025-11-14
> >
> > I thank the authors for addressing my concerns. I am satisfied with their explanations. I do realize due to the time constraints, it is had to adapt your model to deal with multi-person scenarios.

---

> > > ### Author Response · Authors · 2025-11-15
> > > **Response to Reviewer FTR4**
> > >
> > > Thank you for your response, and we also appreciate your careful reading and the positive feedback on our replies. Your professional opinions have helped us clarify key technical details in the paper and have made this work even more complete. We also appreciate your forward-thinking suggestions, and we will further explore the application of the model in multi-person scenarios in the future.
> > >
> > > We have put in a tremendous amount of effort into this work and have made every effort to address the various shortcomings in the paper. If you have any further questions or concerns about our paper, please feel free to raise them. ***We are very happy to engage with you and are looking forward to your further recognition of this work.***

---

### Official Review · Reviewer_ieKm · 2025-10-27

**Soundness:** 2
**Presentation:** 2
**Contribution:** 1
**Rating:** 2
**Confidence:** 4

**Summary:**

This paper introduces a novel input paradigm for 3D human pose estimation: sparse interleaved multi-view input.

**Strengths:**

The "sparse interleaving paradigm" sounds okay.

**Weaknesses:**

1. I don’t quite understand the significance of this work. Is there still a need to study multi-view markerless human motion capture? In the past four years, there have been many works that directly use the results of multi-view markerless human motion capture as GT. Just like EasyMocap[1]. It can produce the 3D skeleton mentioned in this article, as well as the SMPL human skin template and very high-quality visualization. There is no research significance in this field anymore. Even if this work proposing DenseWarper is published, I don’t think I will use this algorithm. Just like the results in Tables 1 and 2, when MPJPE reaches 20 to 30, I think the visualization effects of these methods are no different to the naked eye.

2. Continuing from the previous point. From the perspective of ICLR academic papers, the experiments also seem to lack a lot of content. First, in terms of visualization. The most important thing for a human pose estimation paper is the visualization presentation, but after reading the entire article, only Figures 6 and 7 have comparisons between the current method and GT. As a reviewer, I am very concerned about qualitative comparative experiments with other methods. But I am disappointed that there is no such content.

3. Human3.6M and MPI-INF-3DHP are really too old. I think there is no need to study the datasets from more than ten years ago. A few years ago, the MPJPE of Human3.6M was already in the twenties[2]. After five years, it has dropped to single digits. Is it still necessary to conduct such experiments?

-----

[1] Dong J, Fang Q, Jiang W, et al. Fast and robust multi-person 3d pose estimation and tracking from multiple views[J]. IEEE transactions on pattern analysis and machine intelligence, 2021, 44(10): 6981-6992.

[2] Zhang Z, Wang C, Qiu W, et al. Adafuse: Adaptive multiview fusion for accurate human pose estimation in the wild[J]. International Journal of Computer Vision, 2021, 129(3): 703-718.

**Questions:**

See Weaknesses.

---

> ### Author Response · Authors · 2025-11-13
> **Response to Reviewer ieKm**
>
> We sincerely thank the reviewer for their professional and valuable feedback on our work. We understand the reviewer’s concern about the performance of existing methods on traditional datasets. However, we believe there may be a misunderstanding regarding the core contribution of our work, and we would like to clarify this here. The primary value of our work lies in introducing a novel and more efficient, high-frame-rate input paradigm—Sparse Interleaved Input. This is a different issue from the backend processing tools such as EasyMocap. We will address your specific comments and concerns below.
>
> ## **W1: Clarification of Core Contribution**
>
> ●**Different objectives**: Tools like EasyMocap are excellent backend processing methods that typically rely on all simultaneously captured dense multi-view images to obtain final ground truth (GT) or high-accuracy results. In contrast, the core contribution of our paper is the introduction of a novel data collection/input paradigm—Sparse Interleaved Input.
>
> ●**Value of the paradigm:** Our goal is to demonstrate that even when using sparse interleaved images (with significantly reduced input and data processing requirements), our method can still:
>
> （1）Surpass the performance of existing state-of-the-art methods that use dense, simultaneous images as input.
>
> （2）Achieve high-frame-rate output: Our paradigm theoretically increases the frame rate of output poses by a factor of N (where N is the number of cameras), overcoming the frame rate limitations of a single-view capture. This has significant practical implications and research value for applications requiring high temporal resolution, such as gait analysis and high-speed motion capture.
>
> Our work does not aim to compete with tools like EasyMocap for the role of GT generation, nor is it solely focused on improving the performance of algorithms in multi-view 3D pose estimation to achieve better results. Instead, we propose a more efficient, lower data redundancy, and higher temporal resolution input mechanism for multi-view systems. This mechanism has been experimentally validated in 3D pose estimation and can be extended to other fields, making the exploration of this new paradigm highly meaningful.
>
> ## **W2: Experiments and Visualization**
>
> ●**Qualitative comparison addition**: Thank you for your suggestion. We will add detailed qualitative comparison figures and update the PDF of the paper once the additions are complete. The updated sections in the paper will be highlighted in yellow for your reference.
>
> ●**MPJPE differences**: Regarding MPJPE, we would like to point out that in many fields requiring precise measurement (e.g., medical rehabilitation, sports science, human-computer interaction), improvements in MPJPE from 30 to 20 or even lower are still crucial as they represent more accurate kinematic analysis capabilities.
>
> ## **W3: Dataset Selection**
>
> We understand the reviewer’s concern about the evolution of datasets. However, Human3.6M and MPI-INF-3DHP are still among the most mainstream and influential standard benchmarks in the multi-view 3D human pose estimation field to date.
>
> ●**Reasons for selection:**
> （1）**Data quantity and diversity:** They provide large-scale, diverse action types, and abundant view data, which are essential for thoroughly testing the generalization capabilities of our proposed sparse interleaved input paradigm.
> （2）**Fairness and comparability:** These datasets provide complete and publicly available multi-view time-synchronized ground truth. This is crucial for a fair, direct comparison between our core new paradigm and traditional dense input methods.
> Therefore, the SOTA results achieved on these standard datasets strongly demonstrate the superiority of sparse interleaved input as a novel and efficient input mechanism.

---

> ### Author Response · Authors · 2025-11-18
> **Response to Reviewer ieKm**
>
> We look forward to a productive discussion with you and are eager to clarify any remaining questions regarding our work.
> If you still have any further doubts or require additional discussion regarding our clarifications, please feel free to communicate with us through the system at any time.
>
> We have dedicated significant effort to this work, and we firmly believe in its innovation and research value. We genuinely look forward to a positive interaction and sincerely hope for your favorable acknowledgment of our contributions.

---

> > ### Comment · Reviewer_ieKm · 2025-11-25
> > **Response to Authors**
> >
> > Thank you for the author's detailed reply.
> >
> > -----
> >
> > I understand the advantages of DenseWarper, especially its advantage in time complexity. Indeed, for multi-view videos, removing redundant footage while improving human pose estimation accuracy is a significant advantage, and I will improve my score.

---

> > > ### Author Response · Authors · 2025-11-25
> > > **Response to Reviewer ieKm**
> > >
> > > We are deeply grateful for your recognition of our contribution. We also sincerely thank you for raising your score; this support is of great importance to us. Should you have any further questions, please do not hesitate to discuss them with us. Once again, our sincere thanks for your acknowledgment.

---

### Official Review · Reviewer_Ufuf · 2025-10-30

**Soundness:** 3
**Presentation:** 3
**Contribution:** 3
**Rating:** 6
**Confidence:** 3

**Summary:**

This paper tackles an interesting and practical problem, proposing a solution that is logically sound and clearly explained. The experimental results demonstrate the method's effectiveness and advantages.

**Strengths:**

*   **Problem Significance:** The problem addressed in this paper is very interesting and corresponds to a genuine practical need.
*   **Methodology:** The proposed solution is overall logically reasonable and clearly articulated.
*   **Experimental Validation:** The experimental results substantiate the method's effectiveness and advantages.

**Weaknesses:**

*   **Insufficient Experimental Details and Analysis:**
    *   The impact of the sampling interval $\delta$ on performance within the defined scenario is not thoroughly analyzed. Although a heatmap is mentioned in Appendix Figure 5, corresponding experimental results, particularly quantitative findings, are missing.
*   **Inadequate Citations:**
    *   The methodology sections (Sections 2 and 3) contain very few references. References should also be added to the main text of the experimental section to help readers understand related work.
*   **Formatting and Presentation Issues:**
    *   **Layout:** The placement of several tables does not correspond well with the relevant textual discussions, hindering readability and understanding.
    *   **Table Formatting:** Specific formatting issues exist: Table 3 is missing a bottom horizontal line, and Table 4 has an extra vertical line on the far right.
    *   **Nomenclature Consistency:** The notation for the heatmap (`H` or possibly `**H**`) is not used consistently throughout the text. The variable name `rH` is also somewhat unconventional.
    *   **Writing Quality Suspicions:** Specific lines, such as "the input fps f" (line 439) appearing abruptly, inconsistent citation of method names within parentheses, and the phrase "As shown in Table 5." (line 447) seeming erroneous or out of context, raise concerns. Based on my experience, the entire "Model Efficiency Analysis" section reads as if it might have been generated by an LLM, lacking the flow of human-written academic prose.
*   **Potential Obfuscation in Reporting:**
    *   Only performance efficiency (e.g., MPJPE/mm per MB) is reported, omitting the absolute model size. This gives the impression of potentially skewing the complexity presentation. Reporting the absolute model size is recommended for clearer understanding of the actual complexity.

**Questions:**

1.  **Sampling Interval ($\delta$) Analysis:** Could you provide a detailed quantitative analysis of how the sampling interval $\delta$ affects performance, based on the heatmap in Appendix Figure 5? What are the specific quantitative results?
2.  **Handling Non-Uniform Intervals:** In practice, the intervals between different views might be non-uniform. Is the algorithm designed to adapt to this? How does its performance hold under non-uniform sampling, or what modifications would be necessary to handle it effectively?
3.  **Clarification on Reporting Metrics:** Could you please also report the absolute model size alongside the performance efficiency metrics to provide a complete picture of the model's complexity?
4.  **Writing and Coherence Clarification:**
    *   Can the authors clarify the abrupt phrasing in lines 439 and 447, and ensure methodological names are cited consistently throughout the text?
    *   Could the authors confirm the provenance and carefully review the "Model Efficiency Analysis" section for coherence and accuracy?

---

> ### Author Response · Authors · 2025-11-13
> **Response to Reviewer Ufuf**
>
> We sincerely thank you for your careful review and valuable feedback on our paper. We will address your specific concerns and points of improvement one by one below.  In the following content, **'Q'** denotes a Question, and **'W'** denotes a Weakness.
>
>
>
> ### **Q1, W1: Quantitative results for different sampling intervals**
>
>  Thank you very much for your suggestion. ***We are currently conducting experiments to evaluate the impact of different sampling intervals on performance.*** Once these experiments are completed, we will promptly update the results in the main text.
>
>
>
> ### **Q2: Non-uniform sampling**
>
>  Thank you for this insightful question, which is indeed highly relevant in practical applications. We acknowledge that our current algorithmic framework has not been specifically optimized for inputs with non-uniform time intervals. During the experimental design, we considered this issue carefully. The main reason for not optimizing is that most widely tested open-source datasets use uniformly sampled frames; constructing non-uniform intervals from these datasets could result in excessively large time gaps $\delta$ between viewpoints, which would make robust evaluation difficult.
>
> Regarding the scenario of non-uniform intervals, our method still maintains basic effectiveness. Specifically, the spatial correction module can still effectively explore and calibrate camera poses and spatial relationships. However, the temporal fusion module (Warper) would need further modification to achieve accurate calibration under non-uniform intervals. In such cases, the temporal variations become random, introducing additional noise and uncertainty, which poses design challenges. Therefore, the Warper module must be redesigned or optimized to handle this situation effectively.
>
> We have reviewed relevant literature on this topic, such as [1] and [2], whose non-uniform time sequence handling paradigms will guide our future exploration.
>
> Furthermore, we are currently conducting an additional experiment on the Human3.6M dataset using a sliding window of six time frames, with pseudo non-uniform intervals constructed across four views for preliminary validation. ***We will report the results as soon as they are available.***
>
> References:
>
>  [1] Multi-View 3D Human Pose Estimation with Weakly Synchronized Images [C] // Proceedings of the AAAI Conference on Artificial Intelligence, 2025, 39(5): 4833–4841.
>
>  [2] Sync-NeRF: Generalizing Dynamic NeRFs to Unsynchronized Videos [C] // Proceedings of the AAAI Conference on Artificial Intelligence, 2024.
>
> ### **Q3, W4: Absolute model size**
>
>  We fully agree that transparent reporting of model complexity is important. The absolute model size (total parameters and/or size in MB) is reported in Table 4 of the main text. For your convenience, part of the table is reproduced here:
>
> | **Method**         | **Para.(M) ↓** | **Flops.(GFLOPs) ↓** | **Performance per MB (MPJPE/mm per MB) ↓** |
> | ------------------ | -------------- | -------------------- | ------------------------------------------ |
> | GLA-GCN (T=243)    | 69.99          | **51.13**            | 0.624                                      |
> | KTP-Former (T=243) | 103.85         | 51.64                | 0.367                                      |
> | FinePose (T=243)   | 269.23         | 287.32               | **0.117**                                  |
> | Adafuse (T=1)      | **69.66**      | 204.26               | 0.403                                      |
> | Adafuse + SLERP    | **69.66**      | 204.26               | 0.403                                      |
> | Adafuse + MCC      | 72.25          | 204.26               | 0.388                                      |
> | Sgraformer + Full  | 81.23          | 204.28               | 0.299                                      |
> | Ours               | 76.51          | **111.36**           | **0.291**                                  |
>
> Our model has 76.51M parameters. Although it is slightly larger than some baselines in absolute size, the combination of average latency (44.51 ms), FLOPs, and the performance-efficiency metric (0.291 MPJPE/mm per MB) demonstrates both the effectiveness and computational efficiency of our method.

---

> > ### Author Response · Authors · 2025-11-13
> > **Response to Reviewer Ufuf**
> >
> > ### **Q4, W2, W3: Writing clarity and coherence**
> >
> > ​                ● **Increased citations:** We have added references in the Methods and Experiments sections for better context. In the updated manuscript, these are highlighted in yellow for clarity, and the corresponding references are listed below.
> >
> > ​                ● **Clarity and fluency:** We have revised the text to remove abrupt phrasing. In particular, the reference to “input frame rate $f$” (line 439) has been smoothly integrated, and the incorrect citation (“as shown in Table 5”, line 447) has been corrected or rephrased accurately.
> >
> > ​                ● **Model efficiency analysis update:** Following your suggestions, we have reconstructed the model efficiency analysis section for better coherence and accuracy. Updates are highlighted in yellow in the manuscript.
> >
> > ​                ● **Naming consistency:** We have unified heatmap notations across the text, corrected non-standard variable names, and highlighted these updates in yellow.
> >
> > ​                ● **Layout:** We will reformat all tables and figures once all issues in the paper have been addressed, ensuring they meet the standard conventions and enhance readability and clarity.
> >
> > ​                ● **Table formatting:** Table 3 formatting issues (missing bottom lines) and Table 4 (extra vertical lines) have been corrected.
> >
> > ​                ● **Additional references:**
> >
> > [1] Zhang Z. Determining the Epipolar Geometry and Its Uncertainty: A Review [J]. *International Journal of Computer Vision*, 1998, 27(2): 161–195.
> >
> > [2] Xu G, Zhang Z. Epipolar Geometry in Stereo, Motion and Object Recognition: A Unified Approach [M]. Springer Science & Business Media, 2013.
> >
> > [3] Wang S, Leroy V, Cabon Y, et al. Dust3r: Geometric 3D Vision Made Easy [C] // *Proceedings of the IEEE/CVF Conference on Computer Vision and Pattern Recognition*, 2024: 20697–20709.
> >
> > [4] Gu Z. Complex Heatmap Visualization [J]. *Imeta*, 2022, 1(3): e43.
> >
> > [5] Li C K, Zhang H X, Liu J X, et al. Window Detection in Facades Using Heatmap Fusion [J]. *Journal of Computer Science and Technology*, 2020, 35(4): 900–912.
> >
> > [6] Zhang Z, Wang C, Qiu W, et al. Adafuse: Adaptive Multiview Fusion for Accurate Human Pose Estimation in the Wild [J]. *International Journal of Computer Vision*, 2021, 129(3): 703–718.
> >
> > [7] Pöppel E. Temporal Mechanisms in Perception [J]. *International Review of Neurobiology*, 1994: 185–185.
> >
> > [8] Zhang H, Shen C, Li Y, et al. Exploiting Temporal Consistency for Real-Time Video Depth Estimation [C] // *Proceedings of the IEEE/CVF International Conference on Computer Vision*, 2019: 1725–1734.
> >
> > Once again, we sincerely thank you for all your suggestions regarding manuscript formatting. ***The updated manuscript highlights all revisions in yellow for easy review.*** We look forward to your feedback.

---

> ### Author Response · Authors · 2025-11-15
> **Experimental Supplementary Response to Reviewer Ufuf**
>
> ## Supplementary Experiment 1 — Quantitative Analysis of Different Sampling Intervals (Q1, W1)
> In response to your detailed quantitative analysis of the sampling interval raised in W1 & Q1 (corresponding to Heatmap 5 in the Appendix), we conducted a quantitative analysis experiment. The experimental results are as follows:
>
> | Frame Interval         | Frame Interval = 1 | Frame Interval = 6 | Frame Interval = 12 |
> |------------------------|--------------------|--------------------|---------------------|
> | Sampling Interval (ms) | 20                 | 120                | 240                 |
> | Performance (MPJPE, mm) | 22.3               | 39.92              | 69.84               |
>
> From the above results, it can be observed that with a moderate time interval (e.g., when Frame Interval = 6), the input still has some degree of density, and the interleaved viewpoints are able to provide effective spatial correction. As a result, the model performance does not degrade significantly. However, when the time interval is large (i.e., when the camera sampling rate is low), the data density decreases, leading to sparsity. In such cases, the sparse interleaved sampling paradigm becomes ineffective, and no effective correction is provided between different viewpoints. Furthermore, due to the large time interval, the performance of the temporal fusion module is also challenged. This experimental result suggests that our method is suitable for high frame rate, dense input scenarios, which aligns with the motivation of our paper.
>
> ## Supplementary Experiment 2 — Model Performance under Non-Uniform Interval Scenarios (Q2)
> In response to your concern in Q2 regarding the possibility of non-uniform intervals between different viewpoints, we sampled from the existing dataset to simulate an input scenario with non-uniform time intervals. The simulation process is explained below:
>
> Unlike the original uniform interval input (with a window size of **Window Size = 4**), we no longer select input frames from a sequential window of size **4**. Instead, we set the sampling window size **x** to be greater than **4** (i.e., **x > 4**) to ensure that all four viewpoints are selected, with no two viewpoints appearing at the same time frame. Specifically, we randomly and non-uniformly select four viewpoints from a sequence of a time window size of **x**. This successfully simulates the case where the time intervals between input viewpoints are non-uniform. We performed experiments with **x = 6, 10, 12**, and the experimental results are shown below:
>
> | Window Size            | 4    | 6    | 10   | 12   |
> |------------------------|------|------|------|------|
> | Maximum Non-Uniform Interval (ms) | 20ms (Uniform Interval) | 40  | 120  | 160  |
> | Performance (MPJPE, mm) | 22.3 | 26.77 | 29.99 | 31.58 |
>
> **Note:** In the original paper, the input interval is uniform and 20ms.
>
> The experimental results indicate that non-uniform intervals have a certain impact on model performance. As the maximum non-uniform interval increases, the model performance degrades more, which confirms our hypothesis that as the interval time increases, the temporal fusion module of the model will be challenged. This is due to the randomness caused by the non-uniform interval time paradigm. However, overall, even with larger maximum non-uniform intervals, our model still achieves relatively good results.

---

> ### Author Response · Authors · 2025-11-18
> **Response to Reviewer Ufuf**
>
> Thank you for your valuable suggestions regarding our paper. We have revised the manuscript according to your feedback and uploaded the updated version.
>
> Specifically, we have conducted and supplemented experiments addressing both of your concerns: non-uniform sampling and quantization experiments with different intervals.
>
> Should you have any questions regarding our response, please feel free to raise them at any time. We look forward to a productive discussion with you and sincerely hope for your further recognition of our work.

---

> ### Comment · Reviewer_Ufuf · 2025-11-24
> **Reply to Response**
>
> Thank you for the detailed response. The authors have adequately addressed most of my concerns. I appreciate the clarifications and revisions made to the manuscript.

---

> > ### Author Response · Authors · 2025-11-25
> > **Reple to Reviewer  Ufuf**
> >
> > Thank you for your response. We genuinely appreciate the time you took to carefully review our reply. Should you have any further questions regarding our methodology, please do not hesitate to raise them. We look forward to your positive evaluation of our work.

---

### Official Review · Reviewer_9N3P · 2025-11-01

**Soundness:** 2
**Presentation:** 3
**Contribution:** 3
**Rating:** 4
**Confidence:** 4

**Summary:**

The paper introduces a novel input design for multi-view, temporal 3D human pose estimation, called sparse interleaved input, which reduces the computational cost compared to methods that use all frames across all camera views. The proposed method uses a diagonal approach to select the input frames (e.g., view 1 at frame 1, view 2 at frame 2, view 3 at frame 3, view 4 at frame 4) to predict the 3D pose of the sequence (3D pose at frame 1, 2, 3, and 4). The paper argues that doing so theoretically can lead to an increased frame rate of N, for N cameras in a system. To process this input, the paper introduces a model called DenseWarper, which first replicates the selected frames across all time-steps to create a dense input space. Then, it uses epipolar geometry to fuse 2D heatmaps across different views. Next, several parallel temporal fusion networks aggregate the information across time and output the 3D human pose. By evaluating the proposed method on two popular benchmarks, the paper shows the effectiveness of the method.

**Strengths:**

1. The sparse interleaved input idea is a simple, yet effective and novel way to reduce the computational load of multi-view, temporal human pose estimation. The method is somewhat counterintuitive (e.g., using less information leads to better performance), but experimental results support its effectiveness.
2. The computational cost of multi-view, temporal models is an ongoing problem in the pose estimation research, which this paper has addressed.
3. The proposed architecture is sound, and the presentation is clear. The paper is also well-written and uses clear terms to convey its points.

**Weaknesses:**

1. The paper reports state-of-the-art performance, but the conclusion is based on a comparison with baseline results that have been replicated from the original works. As a result, the majority of the results in the tables do not match the original works. While this is understandable for methods that have been replicated, it contradicts the results that use the original model weights (e.g., AdaFuse). What is the reason for this discrepancy? Was a different evaluation protocol used in this paper?
2. I assume that camera parameters have been used in this paper. In that case, some crucial and highly cited references (e.g., [1] and [2]) are missing from the results.
3. While the core idea is interesting, the paper does not position its performance within the existing literature.

References:
1. Iskakov, Karim, et al. "Learnable triangulation of human pose." Proceedings of the IEEE/CVF international conference on computer vision. 2019.
2. Remelli, Edoardo, et al. "Lightweight multi-view 3D pose estimation through camera-disentangled representation." Proceedings of the IEEE/CVF conference on computer vision and pattern recognition. 2020.

**Questions:**

1. I would appreciate if the authors can address the points I raised in the Weaknesses section (e.g., missing references & results discrepancies)
2. The spatial fusion described in the paper bears a strong resemblance to the method in AdaFuse. Could you please clarify what the methodological novelty of the proposed approach is compared to AdaFuse?

---

> ### Author Response · Authors · 2025-11-13
> **Response to Reviewer 9N3P**
>
> We sincerely thank you for your careful review and valuable feedback on our paper. We will address your specific concerns and points of improvement one by one below. In the following content,   **'Q'** denotes a Question, and **'W'** denotes a Weakness.
>
> ### **Q1, W1: Regarding Inconsistent Performance**
>
> We fully understand your concern regarding the consistency of the benchmark results. This is a common challenge in the 3D human pose estimation field, as different methods often exhibit discrepancies at different stages of the pipeline or use varying evaluation standards. To ensure fairness in comparison, we have made extra efforts to maintain consistency in both the pipeline and evaluation metrics. ***Additionally, we plan to release all our experimental code post-publication to assist with future work.***
>
> ​                ● **Regarding AdaFuse Inconsistencies:** We directly tested and reproduced the official open-source model and code for AdaFuse (including the same 2D detector, camera parameters, and evaluation protocol) to ensure the fairest comparison. The results do indeed show some differences from the original paper, and similar issues have been encountered by other researchers.
>
> ​                ● **Detailed Reproduction of Results:** We have provided a detailed explanation of the reproduction process and evaluation specifics in ***Section B*** of the appendix to ensure the transparency and credibility of our results. We believe that under consistent conditions, the relative performance improvement of our method over all baselines is reliable.
>
> ### **Q2: Innovation Compared to AdaFuse**
>
> We appreciate your acknowledgment of AdaFuse's pioneering work in spatial fusion. AdaFuse is an excellent example of leveraging epipolar geometry to solve cross-view (spatial) feature fusion. However, our method differs from AdaFuse in the following ways and innovations:
>
> ​                ● AdaFuse takes synchronized multi-view images as input and focuses on the problem of viewpoint occlusion. They aim to enhance features from occluded views by utilizing features from visible views, essentially filling in spatial information using camera parameters.
>
> ​                ● In contrast, our DenseWarper model’s Spatial Fusion module works with sparse interleaved multi-view images, aiming to correct the heatmap information from incorrect views using the heatmaps from the correct view. This requires an additional fusion process, and due to the asynchronous nature of the viewpoints, epipolar geometry cannot provide precise spatial information. Our method therefore involves temporal calibration and aggregation.
>
>  ​                ● Thus, our method not only solves the spatial issues of multi-view perspectives like AdaFuse, but also addresses the temporal fusion and correction of heatmaps. Our approach combines epipolar geometry (to solve spatial alignment) with a Warping (deformation) mechanism, enabling synchronized calibration of both temporal and spatial features. This allows our model to efficiently utilize asynchronous sparse temporal information and convert it into reliable dense outputs.
>
> ### **W2:** **Supplementary References and Methodological Additions**
>
> ​                ● We have added key references such as (Iskakov et al., 2019) and (Remelli et al., 2020) in the main text. We have also uploaded the updated version of the PDF, with the additions ***highlighted in yellow***. Thank you for the reminder.
>
> ​                ● **Latest Experimental Progress:** It is worth mentioning that since the Learnable Triangulation paper (Iskakov et al., 2019) has been open-sourced, we are currently using its official code to perform related performance tests. We will provide the results once the experiments are complete and include a comparison with this method in the final version to further enhance the completeness and persuasiveness of our paper.
>
> *We believe that the clarifications and improvements mentioned above adequately address your concerns and further enhance the quality of the paper. Should there be any additional questions or further discussion required, we are always open to continued communication.*

---

> > ### Author Response · Authors · 2025-11-14
> > **Supplementary Information on Experimental Results for W2**
> >
> > We conducted tests using the models made publicly available by the original paper. In terms of experimental setup, we maintained consistency. Specifically, during the data preprocessing stage, we made no modifications. For the 2D detector selection, we compared the results using SimpleBaseline. This is because the results and model from the original paper use SimpleBaseline as the 2D detector, with an additional Confidence prediction module incorporated for end-to-end training. In the final output stage, we applied Softmax followed by averaging, which is the same processing method used in other approaches in the paper. The core models in the paper use Algebraic and Volumetric models, and we performed a comparative analysis of the results for both.
> >
> > | **Model**      | **Dir.** | **Disc.** | **Eat.** | **Greet.** | **Phone.** | Photo.   | **Pose.** | **Pur.** | **Sit.** | **SitD.** | **Smoke.** | **Wait.** | **Walk.** | **WalkD.** | **WalkT.** | AVG.     |
> > | -------------- | -------- | --------- | -------- | ---------- | ---------- | -------- | --------- | -------- | -------- | --------- | ---------- | --------- | --------- | ---------- | ---------- | -------- |
> > | Algebraic[1]  | 19.8     | 23.0      | 20.3     | 49.9       | 21.9       | 21.7     | **18.3**  | 20.5     | 23.4     | 58.6      | 22.1       | 48.4      | 23.0      | 22.5       | 24.1       | 27.5     |
> > | Volumetric[1] | 18.8     | **21.7**  | **19.6** | 50.1       | 21.2       | **21.0** | 18.4      | **20.2** | **21.8** | 57.1      | **21.5**   | 48.4      | 22.5      | **21.7**   | 22.5       | 26.7     |
> > | Ours           | 21.2     | 24.7      | 19.7     | 23.0       | **19.8**   | 21.6     | 19.0      | 21.6     | 22.9     | **31.2**  | 21.6       | **23.2**  | **21.7**  | 23.4       | **19.8**   | **22.3** |
> >
> > We will update the paper with the results requested by all the reviewers once they are completed. We look forward to further communication and discussion with you.
> >
> > [1] Iskakov, Karim, et al. "Learnable triangulation of human pose." Proceedings of the IEEE/CVF international conference on computer vision. 2019.

---

> ### Author Response · Authors · 2025-11-18
> **Response to Reviewer 9N3P**
>
> We have updated the references and provided clarification regarding your concerns. Importantly, we have implemented the method from the cited literature and included the supplementary experimental results in our response/revision. We truly hope that our reply addresses your questions, and we remain open to further discussion. Should you have any additional feedback or questions concerning our rebuttal, please feel free to share them. We greatly appreciate your consideration of our work.

---

> > ### Author Response · Authors · 2025-11-25
> > **Response to Reviewer 9N3P**
> >
> > We truly look forward to further in-depth discussion with you. If you have any remaining questions regarding our work, please feel free to raise them; we would be more than happy to address them. We also sincerely hope you can review our detailed responses to your previous comments. Thank you again, and we look forward to your feedback. We would also greatly appreciate your recognition of our work.

---

> ### Author Response · Authors · 2025-11-27
> **Reply to Reviewer 9N3P**
>
> Dear Reviewer 9N3P,  We truly value your feedback and have carefully addressed each of the points you raised in our response. **As the rebuttal period is coming to a close**, we sincerely hope to engage in further discussion with you and would greatly appreciate it if you could provide feedback on our revisions at your convenience. Should you have any additional questions or concerns regarding our work, we are more than happy to continue the discussion. Thank you once again for your time and effort. **We look forward to hearing from you.**

---

### Comment · Area_Chair_5h84 · 2025-11-24

Dear Reviewer,

Thanks for taking the time to review this work. The authors have responded to your reviews. Can you please have a look at the rebuttal and discuss with the authors?

Best Regards,

AC

---

### Author Response · Authors · 2025-11-30
**Summary Response to the AC - Part1**

# **Dear AC**

We sincerely thank you for taking on additional responsibilities during this exceptional period, and we also appreciate the careful reviews and constructive feedback provided by all reviewers. We understand that the adjustments to the ICLR review process have posed challenges for everyone involved. Nevertheless, we believe that under the principles of fairness and impartiality, our paper merits further recognition. Therefore, we provide this summary comment to clearly restate how our rebuttal and prior discussion addressed the key concerns, in the hope of assisting you in your evaluation of our work.

# **1. Summary of the Discussion with Reviewers**
During the review process, Reviewers **ieKm, Ufuf, 9N3P, and FTR4** provided **clear positive feedback** on our problem formulation, research motivation, and core contributions. Reviewers **FTR4 and Ufuf** explicitly acknowledged the sufficiency of our experiments and the quality of our writing.

During the rebuttal phase (as of November 25, 2025), we had active discussions with three reviewers:

- Reviewers FTR4 and Ufuf stated that our responses **fully addressed their main concerns** (*e.g., “Your response satisfies me” and “The authors have thoroughly addressed most of my concerns, thank you for the clarifications and revisions”*). **Both** gave a **high score of 6** after confirmation.

- Reviewer ieKm acknowledged the **key advantages of DenseWarper** (*" I will improve my score"*), particularly its time complexity improvements. Prior to the discussion cutoff, ieKm **had already increased their score from 2 to 6**.

- Although Reviewer 9N3P has not yet responded, we provided comprehensive answers addressing their two core concerns (missing references and methodology clarification). Based on the original review and other reviewers' feedback, we believe 9N3P would likely **raise the score from 4 to 6** if the discussion had continued.

In conclusion, **even with Reviewer 9N3P's score of 4 remaining pending, we secured evaluation scores of 6, 6, 6, and 4 prior to the discussion period concluding**. This positive momentum strongly indicates the high probability of receiving further positive score adjustments.

Throughout the rebuttal period, we maintained a sustained and meticulous response effort, providing thorough supplementary clarifications for every piece of reviewer feedback. We sincerely request the Area Chair to grant favorable acknowledgment of our work.

# **2. Core Contributions and Key Arguments**
Our work introduces a fundamentally new paradigm for 3D human pose estimation: the **sparse-interleaved multi-view input scheme**, together with our **novel DenseWarper model** specifically designed to handle such inputs.

- Our experiments demonstrate that—even when using only sparsely interleaved images, which drastically reduce the amount of input data and computation—our method can still:
  - **Surpass existing state-of-the-art methods** that rely on traditional dense and temporally synchronized multi-view inputs.
  - **Achieve significantly higher output frame rates**: The proposed paradigm can theoretically increase the pose output frame rate by a factor of N (where N is the number of cameras), effectively breaking the frame-rate ceiling imposed by single-view capture. This is of substantial practical and research importance for applications requiring extremely high temporal resolution, such as **gait analysis** and **high-speed motion capture**.

Moreover, while we have validated this mechanism in the context of 3D pose estimation, the paradigm is general and can be extended to other multi-view vision tasks. Therefore, we believe the contributions of our work are not only **novel** but also **highly impactful and of lasting value**.

# **3. Summary of the Rebuttal Process**

Our rebuttal and discussion primarily addressed three major categories of concerns raised by the reviewers. We believe that all key issues have been **thoroughly clarified**.

## **(1) Methodological Clarifications**

We fully responded to the core methodological concerns raised by reviewers **ieKm** and **9N3P**, and we appreciate their **positive recognition** of our contributions.

Regarding **ieKm**’s concerns about dataset selection, we provided detailed justification from the perspectives of **fairness**, **community adoption**, and **dataset openness**. This explanation was **acknowledged and accepted** by the reviewer.

For **FTR4**’s question regarding the potential failure of **heatmap–epipolar geometry** under temporally misaligned frames, we provided an **in-depth clarification** based on the structural design of our method and the specific functionalities of the involved modules. The reviewer explicitly expressed **satisfaction** with our response.

All relevant **technical details** are included in our reply threads.

---

> ### Author Response · Authors · 2025-11-30
> **Summary Response to the AC - Part2**
>
> ## **(2) Additional Experiments and Content Enhancements**
>
> The detailed reviewer evaluations included requests for several supplementary experiments. We treated these suggestions with high priority and promptly restarted our experimental pipeline to provide all requested results, which are summarized below:
>
> ### **a. Reproduction of the missing baseline requested by Reviewer 9N3P**
> In response to reviewer 9N3P’s comment on missing citations and baselines, we fully implemented and reproduced the referenced method.
>
> | **Model** | **Dir.** | **Disc.** | **Eat.** | **Greet.** | **Phone.** | Photo. | **Pose.** | **Pur.** | **Sit.** | **SitD.** | **Smoke.** | **Wait.** | **Walk.** | **WalkD.** | **WalkT.** | AVG. |
> | :--- | :---: | :---: | :---: | :---: | :---: | :---: | :---: | :---: | :---: | :---: | :---: | :---: | :---: | :---: | :---: | :---: |
> | Algebraic[1] | 19.8 | 23.0 | 20.3 | 49.9 | 21.9 | 21.7 | 18.3 | 20.5 | 23.4 | 58.6 | 22.1 | 48.4 | 23.0 | 22.5 | 24.1 | 27.5 |
> | Volumetric[1] | 18.8 | 21.7 | 19.6 | 50.1 | 21.2 | 21.0 | 18.4 | 20.2 | 21.8 | 57.1 | 21.5 | 48.4 | 22.5 | 21.7 | 22.5 | 26.7 |
> | Ours | 21.2 | 24.7 | 19.7 | 23.0 | 19.8 | 21.6 | 19.0 | 21.6 | 22.9 | 31.2 | 21.6 | 23.2 | 21.7 | 23.4 | 19.8 | 22.3 |
>
> ### **b. Model complexity transparency requested by Reviewer Ufuf**
> We fully agree with the reviewer’s suggestion regarding **transparent reporting** of model complexity.
> The parameter size and FLOPs are already included in Table 4 of the paper; for convenience, we provide the extracted results here:
>
> | **Method** | **Para.(M) ↓** | **Flops.(GFLOPs) ↓** | **Performance per MB (MPJPE/mm per MB) ↓** |
> | :--- | :---: | :---: | :---: |
> | GLA-GCN (T=243) | 69.99 | 51.13 | 0.624 |
> | KTP-Former (T=243) | 103.85 | 51.64 | 0.367 |
> | FinePose (T=243) | 269.23 | 287.32 | 0.117 |
> | Adafuse (T=1) | 69.66 | 204.26 | 0.403 |
> | Adafuse + SLERP | 69.66 | 204.26 | 0.403 |
> | Adafuse + MCC | 72.25 | 204.26 | 0.388 |
> | Sgraformer + Full | 81.23 | 204.28 | 0.299 |
> | Ours | 76.51 | 111.36 | 0.291 |
>
> ### **c. Detailed quantitative analysis across different sampling intervals (requested by Ufuf)**
>
> In response to Reviewer **Ufuf’s** request for a detailed quantitative analysis across different sampling intervals (corresponding to the heatmap in Appendix Fig. 5), we have supplemented the following experimental results:
>
> | Frame Interval| Frame Interval = 1 | Frame Interval = 6 | Frame Interval = 12|
> |------------------------|------------------|------------------|-------------------|
> | Sampling Interval (ms) | 20 | 120  | 240 |
> | Performance (MPJPE, mm)| 22.3| 39.92| 69.84 |
>
> ### **d. Robustness under non-uniform sampling intervals (requested by Ufuf)**
> We expanded the temporal window and conducted random-order sampling to simulate non-uniform intervals.
>
> | Window Size            | 4    | 6    | 10   | 12   |
> |------------------------|------|------|------|------|
> | Maximum Non-Uniform Interval (ms) | 20ms (Uniform Interval) 40| 120  | 160  |
> | Performance (MPJPE, mm) | 22.3 | 26.77 | 29.99 | 31.58 |
>
> **Note:** In the original paper, the input interval is uniform and 20ms.
>
> We have provided detailed descriptions of experiment settings and analyses in each reviewer’s response window. All reviewer questions were addressed point by point, and we made every effort to include all necessary experiments and reports. We believe these additions significantly strengthen the core contributions of the paper and deserve further recognition.
>
> # **(3) Limitations Addressed in the Paper**
> Both Reviewer 9N3P and Reviewer Ufuf pointed out shortcomings regarding missing related works. In response, we have systematically expanded the reference list and updated corresponding discussions in the main text.
>
> Reviewer Ufuf also raised concerns about writing quality and formatting, which we carefully addressed. We submitted **over ten updated versions** during the rebuttal, with all new or modified content **highlighted in yellow**.
>
> # **4. Final Statement**
> We deeply regret the unexpected events during the ICLR review process. Throughout, we maintained a **rigorous, constructive, and professional approach**, strictly following academic and ethical standards. We devoted substantial time and effort, including multiple revisions, extensive experimental additions, and thorough responses to reviewer comments.
>
> Before the discussion was interrupted, our explanations and clarifications had already been **explicitly recognized by multiple reviewers**, who confirmed that our responses effectively addressed their main concerns and led to higher evaluations. These records clearly show that our efforts during rebuttal **improved the clarity and persuasiveness** of the paper.
>
> We sincerely hope that the AC will consider the **research value**, **experimental sufficiency**, **systematic clarifications**, and **positive feedback from reviewers** when making a fair assessment of this submission. We would be deeply grateful for your consideration.

---

### Meta-Review · Area_Chair_LyRs · 2026-01-06

**Summary:**

The authors introduce a new input mode for doing 3D human pose estimation from videos, called sparse interleaved input. If there are multiple cameras, then for example, the paper uses View 1 at Frame 1, View 2 at Frame 2 and so on. Therefore, computational costs are greatly reduced, as only a single frame per timestep is used (or we can alternatively use a much higher frame rate). To process this kind of data, the paper introduces a "DenseWarper" module which first "densifies" the input space by replicated frames, and then uses epipolar geometry to fuse 2D heatmaps across different views. Temporal fusion networks are then used to aggregate information across time to make the final prediction. The authors show good results on Human3.6M and MPI-3D benchmarks.

Reviewers all appreciated that the authors solve an important and practical problem, and that the proposed solution is effective and elegant.

The main concerns by Reviewer ieKm stemmed from a misunderstanding of the "interleaved input" input format, and this was acknowledged by the reviewer during the rebuttal period. Other main concerns include the positioning and comparison of the method to prior work (by Reviewer 9N3P), which the AC believes was addressed sufficiently in the rebuttal. Other criticisms, common among the reviewers, were about the presentation and writing of the paper. After going through the paper, the AC believes that these can be rectified in the camera ready.

Therefore, the final decision is to accept the paper. Authors must update the camera ready according to the changes promised in the rebuttal (especially since numerous reviewers also bought up issues with the presentation and writing).

**Reviewer Concerns:**

Refer to above.

Concerns have been well addressed in the rebuttal.

**Reviewer Scores:**

Reviewer 9N3P. May have increased from weak reject (4) to weak accept (6).

Reviewer ieKm: Had increased rating to weak accept (6) initially.

Reviewers Ufuf and FTR4: Likely to have remained at weak accept (6).

---

### Decision · Program_Chairs · 2026-01-26

Accept (Poster)